# Natural Polyketides Act as Promising Antifungal Agents

**DOI:** 10.3390/biom13111572

**Published:** 2023-10-24

**Authors:** Li Wang, Hui Lu, Yuanying Jiang

**Affiliations:** Department of Pharmacy, Shanghai Tenth People’s Hospital, School of Medicine, Tongji University, Shanghai 200072, China; 2231257@tongji.edu.cn

**Keywords:** polyketides, antifungal, invasive fungal infections

## Abstract

Invasive fungal infections present a significant risk to human health. The current arsenal of antifungal drugs is hindered by drug resistance, limited antifungal range, inadequate safety profiles, and low oral bioavailability. Consequently, there is an urgent imperative to develop novel antifungal medications for clinical application. This comprehensive review provides a summary of the antifungal properties and mechanisms exhibited by natural polyketides, encompassing macrolide polyethers, polyether polyketides, xanthone polyketides, linear polyketides, hybrid polyketide non-ribosomal peptides, and pyridine derivatives. Investigating natural polyketide compounds and their derivatives has demonstrated their remarkable efficacy and promising clinical application as antifungal agents.

## 1. Introduction

Invasive fungal infections significantly threaten human health, resulting in approximately 1.5 million deaths annually [1,2]. The primary culprits responsible for these fatalities are *Candida*, *Cryptococcus*, and *Aspergillus* species [3,4]. The rise in severe underlying diseases and immunocompromised populations, such as those undergoing hematopoietic stem cell transplantation, organ transplantation, immunosuppressive therapy, acquired immune deficiency syndrome, cancer, advanced age, and preterm birth, has further exacerbated the morbidity and mortality associated with invasive fungal infections [5,6]. The current antifungal agents utilized in clinical settings are associated with drawbacks such as drug resistance, limited bioavailability, nephrotoxicity, and a restricted antifungal spectrum [7]. As a result, there is a pressing demand for developing novel antifungal agents to treat invasive fungal infections.

Among the three primary categories of antifungal medications presently accessible, amphotericin B and caspofungin are classified as polyketide compounds. Amphotericin B, a polyene macrolide polyketide, exhibits a broad spectrum of fungicidal activity against *Candida*, *Aspergillus*, and *Cryptococcus* species [8] and remains a preferred treatment option for severe invasive fungal infections [7]. Caspofungin, a non-ribosomal polyketide derivative, selectively targets β-1,3-glucan synthase and impedes fungal cell wall biogenesis with notable selectivity and biological safety compared to amphotericin B [9]. Amphotericin B and caspofungin, both polyketide compounds, have demonstrated clinical efficacy in treating invasive fungal infections. This suggests that developing polyketide compounds as antifungal drugs shows considerable potential.

Polyketides are synthesized through a series of Claisen decarboxylation condensation reactions, utilizing short-chain acyl starting substrates and extension units, including acetyl-CoA, propionyl-CoA, malonyl-CoA, and methylmalonyl-CoA [10]. Polyketides, categorized as secondary metabolites, demonstrate a broad spectrum of structural diversity and are generated by various organisms, including bacteria, fungi, plants, and animals. The biosynthesis of polyketides involves a sequence of condensation reactions catalyzed by three types of polyketide synthases (PKSs): type I PKSs, type II PKSs, and type III PKSs. Type I PKSs are responsible for the biosynthesis of macrolides and related polyenes [11]. Within the category of type I PKSs, there are two distinct subtypes: modular and iterative. Modular type I PKSs are composed of enzyme complexes containing multiple modules, each consisting of linear domains. Each set of domains is utilized only once during the assembly of polyketides [12]. In contrast, iterative type I PKSs possess a single reusable module, with the domains within this module being reused to catalyze multiple rounds of decarboxylation condensation reactions [13]. Type II PKSs, also called aromatic PKSs, are comprised of multiple distinct proteins that function as enzyme complexes. These complexes facilitate repeating a specific chemical reaction by using reusable domains. Typically employing malonyl-CoA as a substrate, Type II PKSs incrementally add two carbon atoms to the polyketide intermediate following each round of decarboxylation condensation reaction. Subsequently, the polyketide is transformed into an aromatic compound under ketoreductase, an aromatase, and a cyclase. The resulting preliminary aromatic polyketide is further modified by an oxygenase, a glycosyltransferase, and a methyltransferase to yield the ultimate aromatic products [14]. Type II PKSs produce aromatic polyketide compounds, including anthracyclines, anticyclones, aureolic acids, tetracyclines, anthracyclines [14], and polyenes [15]. In contrast to the other two types of PKSs, type III PKSs are comprised of a single protein that directly utilizes simple carboxylic acids as substrates, which are activated by acyl-CoA and do not require acyl carrier protein-activated acyl-CoA. Type III PKSs primarily facilitate the biosynthesis of flavonoids, stilbenes, phenylpropanoids, pyrone-type aromatic polyketides, and resorcinol-type aromatic polyketides [16,17,18]. 

This review provides a comprehensive overview of the antifungal properties and mechanisms exhibited by a range of natural polyketide compounds, encompassing macrolide polyethers, polyether polyketides, xanthone polyketides, linear polyketides, hybrid polyketide nonribosomal peptides, and pyridine derivatives. The potential of these natural polyketide compounds in managing invasive fungal infections appears highly promising.

## 2. Macrolide Polyketides

Macrolide polyketides are mainly synthesized by the Type I PKSs. The structural diversity of these compounds is achieved through variations in starting substrates, extension units, modules, and domains, as well as a series of post-modifications that occur after their release. Macrolide polyketides can form glycosidic bonds with one or more sugar moieties. These compounds are classified based on the number of atoms present in the macrolide ring, which includes 12-membered, 14-membered, 24-membered, 26-membered, 32-membered, 36-membered, and 38-membered variants.

Amphidinins Q (**1**), C (**2**) and E (**3**) (Figure 1), which are 12-membered macrolides, are derived from the endophytic dinoflagellates *Amphidinium* species (2012-7-4A strain) found in the marine acoel flatworm *Amphiscolops* species [19]. Amphidinolide Q (**1**) exhibits antifungal activity against *C. albicans* (MIC = 32 μg/mL). On the other hand, amphidinin C (**2**), which is the open-loop structure of amphidinolide Q (**1**), loses its activity against *C. albicans* but demonstrates antifungal activity against *Aspergillus niger* (MIC = 32 μg/mL). Additionally, amphidinin E (**3**), where the carbonyl group at the C-6 position of amphidinin C (**2**) is replaced by a β-hydroxyl group, enhances the antifungal activity against *A. niger* (MIC = 16 μg/mL) [19] (Table 1). 

Rustmicin (**4**) (Figure 2), a 14-membered macrolide, has been obtained from the cultured broth of *Micromonospora narashinoensis* 980-MC [20]. This compound functions as a potent inhibitor of inositol phosphoceramide synthase, thereby impeding the transfer of inositol phosphoceramide to ceramide in synthesizing sphingolipids in fungi [21]. Rustmicin (**4**) exhibits remarkable antifungal activity against *Cryptococcus neoformans* MY2062, *C. neoformans* ATCC9011, *C. tropicalis* MY1012, and *C. albicans* MY1055, *C. albicans* ATCC90028 with MIC values of 0.0001 μg/mL, 0.063 μg/mL, 0.05 μg/mL, 6.25 μg/mL, and 4 μg/mL [22,23]. Nevertheless, the effective reduction of fungal burdens in the spleen and brain necessitates administering high doses of rustmicin (**4**) in *C. neoformans* infected mice model [21]. The reduced potency of rustmicin (**4**) in vivo is attributed to the acceleration of its conversion to the inactive C-2 isoform, γ-lactone, by serum, which is subsequently degraded [21]. Additionally, the antifungal activity of rustmicin (**4**) is diminished due to the conversion of its enol ether structure at the C-6 position to an inactive ketone structure under acidic conditions. However, substituting the methoxy group at the C-6 position with methylthio (**5**) (Figure 2) exhibits weaker antifungal activity against *C. neoformans* ATCC9011 (MIC = 0.5 μg/mL) and *C. albicans* ATCC90028 (MIC = 64 μg/mL) [23] (Table 1). Compared to the oxygen atom, the sulfur atom exhibits lower electronegativity and cannot form hydrogen bonds. Consequently, sulfur is not an efficient hydrogen bond donor compared to oxygen. Replacing the original oxygen atom with a sulfur atom elevates the logP value and augments the lipophilicity of compound **5**, consequently leading to diminished water solubility.

In substituting the methoxy group at the C-6 position with hydrogen (**6**), fluoroalkoxy (**7**), and halogen (**8**) (Figure 2), the antifungal activity of the compounds is greatly reduced or even lost [23]. Compound **6** loss had a significant effect on *C. neoformans* ATCC9011 (MIC > 64 μg/mL) and *C. albicans* ATCC90028 (MIC > 64 μg/mL). Compound **7**, bearing a fluoroalkoxy group, also belongs to the enol ether structure and has altered acid-base stability and lipophilicity. Although bearing a fluoroalkoxy group can enhance the stability of compound **7**, it will reduce the pH value and weaken the alkalinity. Meanwhile, compared with the methyl group, the logP value of a fluoroalkoxy group increased, and its solubility in water decreased. Therefore, compound **7** exhibited attenuated antifungal activity against *C. neoformans* ATCC9011 (MIC = 32 μg/mL) and lost its activity against *C. albicans* ATCC90028 (MIC > 64 μg/mL). Compound **8** exhibited attenuated antifungal activity against *C. neoformans* ATCC9011 (MIC = 16 μg/mL) and lost its activity against *C. albicans* ATCC90028 (MIC > 64 μg/mL). Removing the methoxy group from the C-6 position to the C-5 position (**9** and **10**) (Figure 2), the antifungal activity of these compounds also disappeared (MIC > 64 μg/mL) [23] (Table 1). Evidence supports the idea that the enol ether structure at position C-6 is essential for the antifungal activity of rustmicin. 21-Hydroxyrustmicin (**11**) (Figure 2) is isolated from *Micromonospora* sp. UV Mutant (MA 7186) and exhibits stronger antifungal activity (MIC = 0.024 μg/mL) against *C. tropicalis* MY1012 and weaker antifungal activity against *C. albicans* MY1055 (MIC = 12.5 μg/mL) and *C. neoformans* MY2062 (MIC = 0.1 μg/mL) [22]. Galbonolide B (**12**) (Figure 2) is isolated from *Micromonospora* species (MA 7094) and UV Mutant (MA 7186) [22]. Galbonolide B (**12**), substituting the methoxy group at the C-6 position with a methyl group, greatly reduces antifungal activity against *C. neoformans* MY2062 (MIC = 12.5 μg/mL), *C. tropicalis* MY1012 (MIC = 200 μg/mL), and *C. albicans* MY1055 (MIC > 200 μg/mL). 21-Hydroxygalbonolide B (**13**) (Figure 2) is isolated from *Micromonospora* species UV Mutant (MA 7186) greatly enhances the antifungal activity of galbonolide B (**12**) against *C. tropicalis* MY1012 and *C. neoformans* MY2062 with MIC values of 0.78 μg/mL and 3.1 μg/mL, respectively [22] (Table 1). 

Preussolides A (**14**) and B (**15**) (Figure 3), 24-membered macrolides, have been extracted from the coprophilous isolates of *Preussia typharum* [24]. These compounds are characterized by a distinctive phosphoethanolamine substituent, with the only difference being the presence or absence of a double bond between the C-10 and C-11 positions. Preussolide B (**15**), with the double bond, exhibits weak antifungal activity against *C. neoformans* H99 (37 °C) (MIC = 32 μg/mL), *C. neoformans* H99 (23 °C) (MIC = 32 μg/mL), and *C. albicans* ATCC 10231 (MIC = 256 μg/mL). On the other hand, preussolide A (**14**), lacking the double bond, shows enhanced activity against *C. neoformans* H99 (37 °C) (MIC = 4 μg/mL), *C. neoformans* H99 (23 °C) (MIC = 8 μg/mL), *C. albicans* ATCC 10231 (MIC = 256 μg/mL) and *A. fumigatus* AF239 (MIC = 8 μg/mL) [24] (Table 1). Preussolide A (**14**) exhibits stronger activity, maybe due to the C10-C11 single bond being more stable than the C10-C11 double bond. Additionally, single and double bonds can change the stereoconfiguration of compounds. 

Oligomycin A, a 26-membered macrolide with a bicyclic spiroketal, is isolated from *Streptomyces* [25]. Oligomycin A presents potently broad spectrum antifungal activity against *C. albicans* ATCC 24433 (MIC = 2–4 μg/mL), *Candida krusei* 432M (MIC = 1–2 μg/mL), *Candida parapsilosis* ATCC 22019 (MIC = 2 μg/mL), *Candida utilis* 84 (MIC = 1 μg/mL), *Candida tropicalis* 3019 (MIC = 1 μg/mL), *A. niger* (MIC = 0.5–2 μg/mL), *Cryptococcus humicolus* ATCC 9949 (MIC = 2 μg/mL), and *Trichophyton mentagrophytes* ATCC 9533 (MIC = 10 μg/mL) [26,27,28,29,30]. The oligomycin A molecule contains a C16-C19 diene system that can adopt two conformations, namely *s-Trans* (**16**) and *s-Cis* (**17**) (Figure 4). When the *s*-*Cis* conformation of the diene system in oligomycin A is replaced with benzo-quinone (**18**) and N-benzyl maleimide (**19** and **20**) (Figure 4), these resulting cycloadducts exhibit a loss of activity against *C. albicans* ATCC 24433, *C. parapsilosis* ATCC 22019, *C. krusei* 432M, and *A. niger* 137a (MICs > 32 μg/mL). This activity loss is likely attributed to the reduced permeability of the cycloadducts across the fungal cell wall [26]. (33S)-oligomycin A (**21**) (Figure 4), substituted the α-hydroxy group at the C-33 position with a β-hydroxy group, exhibits comparable antifungal activity against *C. parapsilosis* ATCC 22019 (MIC = 1 μg/mL), *C. albicans* ATCC 24433 (MIC = 4 μg/mL), *C. utilis* 84 (MIC = 2 μg/mL), *C. tropicalis* 3019 (MIC = 1 μg/mL), *C. krusei* 432 M (MIC =4 μg/mL), and *A. niger* 137 a (MIC = 2 μg/mL) to oligomycin A [28]. The 8, 9 carbon bond of oligomycin A is disrupted to obtain the acyclic derivative (**22**) (Figure 4) of oligomycin A, and the antifungal activity of these acyclic compounds against *C. albicans* ATCC 14053 (MIC = 16 μg/mL) and *A. niger* ATCC 16404 (MIC = 4 μg/mL) is greatly reduced due to losing the inhibitory activity against F_0_F_1_ ATP synthase-containing proteoliposomes [31]. Bromo-oligomycin A (**23**) (Figure 4), in which the tetrahydropyrane ring contains bromine in the C-16 position, loses antifungal activity against *A. niger* ATCC 16404 (MIC >16 μg/mL), *C. albicans* ATCC 14053 (MIC > 16 μg/mL) except for *C. humicolus* ATCC 9949 (MIC = 2 μg/mL) may be explained that the derivative forms a webbed structure that distorts conformation in comparison with oligomycin A, and the derivative lacks the free 13-OH group that interacts with the target site [29]. Oligomycin E (**24**) (Figure 4), the analog of oligomycin A *s-Cis* (**17**), is extracted from *Streptomyces* species strain HG29 isolated from Saharan soil [32]. Oligomycin E (**24**), in which an α-methyl group replaces the β-methyl group at the C-4 position of oligomycin A and the bicyclic spiroketal has additional hydroxyl and carbonyl groups, exhibits comparable antifungal activity against *Aspergillus carbonarius* M333 (MIC = 2 μg/mL), *A. westerdijkiae* NRRL 3174 (MIC = 8 μg/mL), *A. parasiticus* CBS 100926 (MIC = 4 μg/mL), *A. nidulans* KE202 (MIC = 75 μg/mL), *A. niger* OT304 (MIC = 4 μg/mL), *A. terreus* CT290 (MIC = 75 μg/mL), and *A. fumigatus* CF140 (MIC = 100 μg/mL) in vitro [32]. Oligomycin C (**25**) (Figure 4), the analog of oligomycin A *s-Trans* (**16**), is extracted from *Streptomyces diastaticus* [30]. Oligomycin C (**25**), in which the β-hydrogen replaces the β-hydroxyl group at the C-12 position of oligomycin A, exhibits comparable antifungal activity against *A. niger* ATCC 10335 (MIC = 2 μg/mL) [30]. Oligomycin A annelates the structure of nitrone and forms a cyclic nitrone to form a new compound **26** (Figure 4), which reduces the cytotoxicity but loses the antifungal activity [33]. Compounds **27** and **28** (Figure 4), formed by oligomycin A linked to pyrazolo [1,5-a] pyridine, show reduced cytotoxicity and weaker antifungal potential against *A. niger* (MIC = 2 μg/mL) to oligomycin A (MIC = 0.125 μg/mL) [33]. Neomaclafungins A-I (**29**–**37**) (Figure 4), homologs of oligomycin A, have been extracted from the fermentation broth of *Actinoalloteichus* species NPS702 [27]. These compounds exhibit stronger antifungal activity against *T. mentagrophytes* ATCC 9533, with MIC values ranging from 1 to 3 μg/mL, which may be explained by the absence of the ketones in the 26-membered ring and the different substituent ation at the C-24 position [27] (Table 1).

Brasilinolides A (**38**) and B (**39**) (Figure 5), 32-membered macrolides with a tetrahydropyran ring and a 2-deoxyfucopyranose, are obtained through the fermentation of *Nocardia brasiliensis* IFM0406 [34,35]. Brasilinolide A (**38**) exhibited selective inhibition of *A. niger* IFM 40406 (MIC = 3.13 μg/mL) [34]. The malonyl side chain of brasilinolide A (**38**) is not essential for its antifungal activity [36]. Brasilinolide B (**39**), changing the malonyl side chain and the sugar moiety of brasilinolide A (**38**), has broad-spectrum antifungal activity against *A. niger* (MIC = 12.5 μg/mL), *A. fumigatus* IFM 41219 (MIC = 12.5 μg/mL), *C. albicans* ATCC 90028 (MIC = 25 μg/mL), *C. albicans* IFM 40007 (MIC = 12.5 μg/mL), *C. albicans* 94–2530 (MIC = 25 μg/mL), *C. krusei* M 1005 (MIC = 25 μg/mL), *C. parapsilosis* ATCC 90018 (MIC = 12.5 μg/mL), *C. glabrata* ATCC 90030 (MIC = 25 μg/mL), *C. neoformans* ATCC 90112 (MIC = 12.5 μg/mL), and *C. neoformans* 145 A (MIC = 25 μg/mL) [35]. Compared with brassinolide A (**38**), copiamycin (**40**) (Figure 5) replaces the malonyl side chain from the C-23 position to the C-21 position without the sugar moiety, and copiamycin shows antifungal activity against *C. albicans* Yu 1200 (MIC = 25 μg/mL) [36]. Methylcopiamycin (**41**) (Figure 5), the 15-OH methylation product of copiamycin, has the same antifungal activity against *C. albicans* Yu 1200 (MIC = 25 μg/mL) [36]. Demalonylmethylcopiamycin (**42**) (Figure 5), removing the malonyl side chain connected to C-21 of methylcopiamycin, shows stronger antifungal activity against *C. albicans* Yu 1200 (MIC = 6.25 μg/mL) compared to copiamycin and methylcopiamycin [36] (Table 1). Langkolide (**43**) (Figure 5), a compound obtained from the mycelium of *Streptomyces* species Acta 3062, replacing the malonyl side chain connected to C-23 of brassinolide A (**38**) with acetyl side chain connected to C-23 position and having disaccharide moiety with 1,4-naphthoquinone connected to C-37 of the aglycone moiety, has been found to exhibit inhibitory effects on the growth of *Candida glabrata* and *C. albicans*, with half maximal inhibitory concentration (IC_50_) values of 1.00 ± 0.02 and 1.23 ± 0.10 μM, respectively [37].

The structures of 1,4-naphthoquinone, glycosylated moiety, and large polyol macrolide aglycones are important in cyphomycin and its derivatives, which are 36-membered macrolides [38]. Cyphomycin (**44**) (Figure 6), isolated from the microbiome of the fungus-growing ant *Cyphomyrmex* species, a Brazilian *Streptomyces* ISID311, exhibits potent antifungal activity against resistant triazole *A. fumigatus* 11628 (MIC = 0.5 μg/mL), resistant echinocandin *C. glabrata* 4720 (MIC = 0.5 μg/mL), and resistant triazole, echinocandin, and amphotericin B *C. auris* B11211 (MIC = 4 μg/mL) [38]. Furthermore, cyphomycin has demonstrated in vivo antifungal efficacy [38]. The neutropenic mouse model of candidiasis is subjected to treatment with cyphomycin, resulting in attenuation of the renal fungal burden in mice compared to the zero-hour control [38]. Caniferolides (**45**), B (**46**), C (**47**), and D (**48**) (Figure 6) are isolated from the marine *Streptomyces caniferus* CA-271066 [39]. Caniferolide C (**47**) shows antifungal activity against *A. fumigatus* ATCC46645 (MIC = 4–8 μg/mL) and *C. albicans* MY1005 (MIC = 0.5–1 μg/mL). Caniferolide C (**47**) is a compound that converts the C32-C33 double bond of cyphomycin to an epoxy group. Caniferolide B (**46**) shows antifungal activity against *A. fumigatus* ATCC46645 (MIC = 2–4 μg/mL) and *C. albicans* MY1005 (MIC = 1–2 μg/mL). Caniferolide B (**46**) is a compound that adds a hydroxyl group to C-18 of caniferolide C. Caniferolide A (**45**) shows antifungal activity against *A. fumigatus* ATCC46645 (MIC = 2–4 μg/mL) and *C. albicans* MY1005 (MIC = 0.5–1 μg/mL). The hydroxyl group linked to C-19 of caniferolide B is replaced by a sulfate ester group to give the compound caniferolide A. Caniferolide D shows antifungal activity against *A. fumigatus* ATCC46645 (MIC = 4–8 μg/mL) and *C. albicans* MY1005 (MIC = 0.5–1 μg/mL). Unlike caniferolide A, Caniferolide D has no hydroxyl group at the C-15 position. Iseolides A (**49**), B (**50**), and C (**51**) (Figure 6) identified from the culture extract of *Streptomyces* species DC4-5, isolated from a stony coral *Dendrophyllia*, exhibits potent antifungal activity against *C. albicans* NBRC0197 with the MIC values of 0.39 μg/mL, 6.25 μg/mL, and 3.16 μg/mL [40]. Iseolide B and cyphomycin differ in that a deoxysugar is attached at C-52 of the sugar moiety, and iseolide B shows antifungal activity against *C. albicans* with MIC value of 6.25 μg/mL [40]. Iseolide A (**49**), which links an α-hydroxyl group at the C-18 position of the aglycone of Iseolide B (**50**), shows 16 times higher antifungal activity (MIC = 0.39 μg/mL) than iseolide B [40]. Iseolide C (**51**) does not attach to anything but contains this methyl group, which has greatly reduced antifungal activity (MIC = 3.16 μg/mL) compared to iseolide A (**49**) [40]. Astolides A (**52**) and B (**53**) (Figure 6) are isolated from *Streptomyces hygroscopicus* and are collected from alkaline soil in the Saratov region of Russia [41]. Astolide A (**52**), removing the α-methyl group at position C-4 of the aglycone moiety of iseolide A (**49**), is found to be effective for *C. albicans* ATCC 14053 (MIC = 2.5 μg/mL), *A. niger* ATCC 16404 (MIC = 1.25 μg/mL), *C. albicans* 1582 (MIC = 2.53 μg/mL), *C. tropicales* 1402 (MIC = 5.06 μg/mL), and *A. niger* 219 (MIC = 2.53 μg/mL) [41]. Astolide B (**53**), adding a hydroxyl group at the C-3” position of the deoxysugar of astolide A (**52**), shows enhanced antifungal activity against *C. albicans* ATCC 14053 (MIC = 1.25 μg/mL), *A. niger* ATCC 16404 (MIC = 0.6 μg/mL), *C. albicans* 1582 (MIC = 2.51 μg/mL), *C. tropicales* 1402 (MIC = 5.01 μg/mL), and *A. niger* 219 (MIC = 2.51 μg/mL) compared with astolide A (**52**) [41] (Table 1).

Guanidylfungin A (**54**) (Figure 7), a 36-membered macrolide, is isolated from the mycelia of *Streptomyces hygroscopicus* No. 662 and exhibits antifungal activity against *C. albicans* IAM 4888 (MIC = 12.5 μg/mL), *C. albicans* Yu 1200 (MIC = 50 μg/mL), and *A. fumigatus* IAM 2153 (MIC = 25 μg/mL) [36,42]. Structural modification of guanidylfungin A gives alkyl products, including methylguanidylfungin A (**55**), ethylguanidylfungin A (**56**), butylguanidylfungin A (**57**), and allylguanidylfungin A (**58**) (Figure 7), the activity of these compounds is comparable to or slightly lower than that of guanidylfungin A (**54**) [36]. Methylguanidylfungin A (**55**) shows antifungal activity against *C. albicans* IAM 4888 (MIC = 25 μg/mL), *C. albicans* Yu 1200 (MIC = 50 μg/mL), and *A. fumigatus* IAM 2153 (MIC = 12.5 μg/mL). Ethylguanidylfungin A (**56**) shows antifungal activity against *C. albicans* IAM 4888 (MIC = 25 μg/mL) and *A. fumigatus* IAM 2153 (MIC = 25 μg/mL). Butylguanidylfungin A (**57**) shows antifungal activity against *C. albicans* IAM 4888 (MIC = 25 μg/mL) and *A. fumigatus* IAM 2153 (MIC = 50 μg/mL). Allylguanidylfungin A (**58**) shows antifungal activity against *C. albicans* IAM 4888 (MIC = 25 μg/mL) and *A. fumigatus* IAM 2153 (MIC = 25 μg/mL). Compound **59** (Figure 7), removing the malonyl side chain at the C-23 position of methylguanidylfungin A (**55**), increases antifungal activity against *C. albicans* IAM 4888 (MIC = 3.12 μg/mL), *C. albicans* Yu 1200 (MIC = 6.25 μg/mL), and *A. fumigatus* IAM 2153 (MIC = 3.12 μg/mL) due to increased solubility in water [36]. Compound **60** (Figure 7), the ring-opening structure of the tetrahydropyran ring of guanidylfungin A (**54**), loses antifungal activity against *C. albicans* IAM 4888 (MIC > 100 μg/mL), *C. albicans* Yu 1200 (MIC > 100 μg/mL), and *A. fumigatus* IAM 2153 (MIC = 50 μg/mL). Compound **61** (Figure 7), removing the malonyl side chain at the C-23 position of compound **60**, cannot restore antifungal activity against *C. albicans* IAM 4888 (MIC = 100 μg/mL), *C. albicans* Yu 1200 (MIC = 100 μg/mL), and *A. fumigatus* IAM 2153 (MIC = 12.5 μg/mL) despite the increased water solubility [36] (Table 1). These lines of evidence suggest that the tetrahydropyran ring is necessary for guanidylfungin A (**54**) activity, but the malonyl group is not.

In the 1950s, amphotericin B (**62**) (Figure 8), a 38-membered macrolide isolated from *Streptomyces nodosus*, was introduced to the clinic as a natural antifungal agent, demonstrating broad-spectrum antifungal activity against various invasive fungi [7,8] (Table 1). Its mechanism of action involves acting as a “sterol sponge”, forming aggregates outside the cell membrane to extract ergosterol from the bilayer and kill yeasts [43]. However, amphotericin B (**62**) exhibits dose-dependent renal and hematopoietic toxicity by targeting and extracting cholesterol from host cell membranes, damaging host cells [44]. The introduction of lipid-based formulations of amphotericin B (**62**) during the mid-1990s significantly reduced its nephrotoxicity [44]. However, the clinical application of amphotericin B (**62**) is restricted due to its requirement for intravenous administration. A new nanoparticle crystal encapsulated formulation of amphotericin B (**62**) called cochleated amphotericin B (CAmB) is a novel oral formulation of amphotericin B [45]. CAmB demonstrates in vitro activity against *C. neoformans*, *Candida* species, and *A. fumigatus* [46]. Intraperitoneal injection of CAmB significantly increases the survival rate of mice infected with *C. albicans* [47]. Using a systemic *aspergillosis* model, survival is 70% after 14 days at oral doses of 20 mg/kg and 40 mg/kg of CAmB and the fungal burden of lung, liver and kidney is reduced by more than 100 times [48]. Using a 3-day delayed model of murine *cryptococcal* meningoencephalitis and a large inoculum of a highly virulent strain of serotype A C. *neoformans*, CAmB, in combination with flucytosine, is found to have efficacy equivalent to parental amphotericin B deoxycholate with flucytosine and superior to oral fluconazole without untoward toxicity [49]. In a Phase I trial, the safety and tolerability of CAmB treatment for *cryptococcal* meningitis in HIV-infected patients were assessed, revealing that oral CAmB was well tolerated and not nephrotoxic when compared to intravenous CAmB (NCT04031833) [46]. Furthermore, a Phase II trial examined the efficacy of CAmB in patients with azole-resistant chronic mucocutaneous *candidiasis*, and subsequent clinical trials demonstrated favorable tolerance and safety outcomes (NCT02629419) [50].

## 3. Polyether Polyketides

Type I PKSs catalyze the decarboxylation reaction of the substrate to generate a polyketide skeleton. This skeleton then is undergone a series of post-modifications, including epoxidation, epoxide ring opening cascade to ether, hydroxylation, methylation, and glycosylation, ultimately forming polyether polyketides. Polyether polyketides are natural polyketide products with multiple asymmetric centers and two or more tetrahydrofuran and tetrahydropyran rings. Polyether polyketides can be categorized into three groups based on their distinct chemical structures: polyethers with long-chain and multi-hydroxyl groups, macrolide polyethers, and ladder-like polyethers.

### 3.1. Polyethers with Long-Chain and Multi-Hydroxyl Groups

Polyethers with long-chain and multi-hydroxyl groups are water-soluble polyethers whose linear polyketide skeletons only partially form ether rings, and the ether rings and polyketide skeletons are highly hydroxylated.

Amphidinols possess a bis-tetrahydropyran parent structure linked by a C6 alkyl chain with a hydrophilic polyhydroxyl chain and a hydrophobic polyene tail. The hairpin-shaped structure of amphidinols is believed to be crucial for their antifungal activity [51,52]. Amphidinols exhibit potent antifungal activity by interacting with the membrane integration protein glycophorin A, thereby increasing cell membrane permeability [53]. Amphidinol 3 (**63**) (Figure 9) is isolated from a marine dinoflagellate, *Amphidinium klebsii* [53]. Amphidinol 3 (**63**) has antifungal activity against *A. niger* by disk diffusion method with a minimum effective concentration (MEC) of 8 μg/disk [52]. The presence of the C1 to C20 polyol moiety in amphidinol 3 (**63**) does not contribute to its antifungal activity, while the C21 to C30 moiety is crucial for its antifungal activity [52]. When the C21-C67 section (**64**) (Figure 9) of amphidinol 3 (**63**) is retained, it exhibits a MEC of 20 μg/disk against *A. niger*. However, when only the C31-C67 moiety (**65**) (Figure 9) of amphidinol 3 (**63**) is retained, the compound loses its activity against *A. niger* [52]. The hydrophobic polyene tail of amphidinol 3 (**63**) inserts into the lipid bilayer membrane and the hairclip amphidinol 3 (**63**) results in pore formation that increases the permeability of the cell membrane in a sterol-dependent manner [52]. Amphidinol 3 (**63**) is believed to exhibit two distinct mechanisms of action: the barrel stave model and the toroidal model. According to the barrel stave model, amphidinol 3 (**63**) permeates the lipid bilayer directly, forming pores. Conversely, the toroidal model involves the insertion of the hydrophobic polyene tail of amphidinol 3 (**63**) into the lipid bilayer, while the hydrophilic polyol portion interacts with the cell membrane surface, ultimately leading to pore formation [52]. Amphidinol 18 (**66**) (Figure 9) is isolated from a marine dinoflagellate, *Amphidinium carterae* [54]. Amphidinol 18 (**66**), which has a carbonyl group added to the polyol structure compared to amphidinol 3 (**63**), shows antifungal activity against *C. albicans* with MIC values of 9 μg/mL [54]. Amphidinol A (**67**) (Figure 9) is isolated from a marine dinoflagellate, *Amphidinium carterae* [51]. Amphidinol A (**67**), replacing the C54-C63 polyene chain of amphidinol 18 (**66**) with alkyl chains, exhibits lower antifungal activity against *C. albicans*, with MIC values of 19 μg/mL [51]. Karatungiol A (**68**) (Figure 9) is isolated from marine dinoflagellates [55]. Karatungiol A (**68**), replaced with a longer polyol moiety than amphidinol A (**67**), exhibits potent antifungal activity against *A. niger* [55]. Amphidinol 6 (**69**) (Figure 9) is isolated from a marine dinoflagellate, *Amphidinium klebsii* [53]. Amphidinol 6 (**69**), which has a longer polyol moiety and a shorter polyene tail than amphidinol 3 (**63**), exhibits similar antifungal activity against *A. niger* with a MEC value of 6 μg/disk [53]. Amphidinols 2 (**70**) and 7 (**71**) (Figure 9) are isolated from a marine dinoflagellate, *Amphidinium klebsii* [53]. Amphidinols 2 (**70**) and 7 (**71**), having shorter polyol moieties than amphidinol 6 (**69**), show similar antifungal activity against *A. niger* with MEC values of 6 and 10 μg/disk [53]. Desulfurization amphidinol 7 (**72**) (Figure 9), where the 2-hydroxyl group replaces the sodium sulfonate of amphidinol 7, exhibits stronger antifungal activity against *A. niger* with a MEC value of 8 μg/disk [53]. Amphidinols 20 (**73**) and 21 (**74**) (Figure 9) are isolated from a marine dinoflagellate, *Amphidinium carterae* [56]. Amphidinols 20 (**73**) and 21 (**74**), having longer polyol moieties than amphidinol 6 (**69**), exhibit negligible antifungal activity against *A. niger* [56]. This could be attributed to the elongated polyol chains, which hinder their integration into the lipid bilayer, disrupting the barrel stave model and diminishing their membrane destruction activity [56]. Another possible explanation for the diminished bioactivity of amphidinols 20 (**73**) and 21 (**74**) is their increased solubility, which hinders the polyol from attaching to the membrane surface [56]. Carteraol E (**75**) (Figure 9) is isolated from marine dinoflagellates [57]. Carteraol E (**75**), having a different polyol moiety compared to amphidinol 6 (**69**), exhibits lower antifungal activity against *A. niger* with an MEC value of 15 μg/disk [57] (Table 1).

### 3.2. Ladder-like Polyethers

Ladder-like polyethers are composed of ether rings, which are composed mainly of six-membered rings. The ether rings are arranged into ladder-like structures by trans configuration. The oxygen atoms of the adjacent ether rings are alternately located at the upper and lower ends of the ring. Ladder-like polyethers have low polarity and are lipid-soluble compounds.

Yessotoxin (**76**) (Figure 10), a compound derived from the dinoflagellate *Protoceratium reticulatum* found in Mutsu Bay, Japan, has been investigated regarding its structure–activity relationship (SAR) [58]. Desulfated yessotoxin (**77**) and hydrogen-desulfated yessotoxin (**78**) (Figure 10), two derivatives of yessotoxin, have been synthesized for this purpose [58]. Desulfated yessotoxin (**77**) has been found to exhibit reduced hydrophilicity and increased antifungal activity against *A. niger* [58] (Table 1). Hydrogen-desulfated yessotoxin (**78**) is a product of the hydrogenation of the polyene side chain of desulfated yessotoxin (**77**), and its antifungal activity is comparable to that of desulfated yessotoxin (**77**), indicating that the ladder-shaped polyether structure, as opposed to the polyene side chain, is critical for the antifungal activity of yessotoxin. Desulfated yessotoxin (**77**) has been found to bind to the transmembrane α-helix motif of the membrane integral protein glycophorin A, thereby inducing the dissociation of glycophorin A oligomers into dimers and monomers [58]. Despite its antifungal activity, yessotoxin (**76**) has been observed to induce subacute cardiotoxicity [59]. In vitro studies have shown that human ether-a-go-go related gene (hERG) Chinese hamster ovary cells treated with 100 nM yessotoxin for 12 or 24 h exhibit increased hERG potassium channels on the cell surface [59]. In vivo experimentation involves the intraperitoneal injection of rats with either 50 μg/kg or 70 μg/kg yessotoxin (**76**) every 4 days, resulting in significant physiological changes such as bradycardia, hypotension, cardiac structural alterations, and elevated levels of plasma tissue metalloproteinase-1 inhibitor after 15 days [59]. Additional studies on its structure–activity relationship are warranted to improve the antifungal efficacy of yessotoxin (**76**) while mitigating its cardiotoxicity.

### 3.3. Macrolide Polyethers

Macrolide polyethers are end-to-end polyether products in the form of ester bonds. Forazoline A (**79**) (Figure 11) is obtained from *Actinomadura* species strain WMMB-499 isolated from the ascidian Ecteinascidia turbinate [60]. Forazoline A (**79**) exhibits favorable water solubility, with a concentration of approximately 5 mg/mL [60]. The chemogenomic approach suggests that forazoline A (**79**) may interfere with the integrity of the cell membrane by disrupting phospholipid homeostasis [60]. Forazoline A (**79**) exhibits growth inhibition of *C. albicans* K1 with a MIC value of 16 μg/mL [60] (Table 1). In a mouse model of *C. albicans* infection, forazoline A (**79**) demonstrates comparable in vivo efficacy to amphotericin B (**62**) without toxicity [60]. The administration of forazoline A (**79**) at a dose of 0.125 mg/kg reduced the colony-forming unit more than 10 times in the fungal burden of mice kidneys after 8 h, compared to the control group [60].

## 4. Xanthone Polyketides

Xanthone is synthesized through different pathways in plants, fungi, and lichens. In plants, it is synthesized via the shikimate and acetate pathways, while in fungi and lichens, it is synthesized through the polyketide pathway. This review focuses on the biosynthesis of xanthone polyketides through the polyketide pathway. The substrates, such as acetyl-CoA and propionyl-CoA, undergo decarboxylation and condensation to form polyketides. Ketoreductase, aromatases, and cyclases then catalyze these polyketides to form aromatic polyketides. Finally, post-modification processes lead to the formation of xanthone polyketides. 

Xanthone is an aromatic oxygenated heterocyclic molecule with a dibenzo-γ-pirone scaffold [61]. Xanthones can be categorized into three distinct structural groups: simple xanthones, O-heterocyclic xanthones, and polycyclic xanthones. Simple xanthones are characterized by hydroxy, methyl, carboxyl, or methoxy substitutions. O-heterocyclic xanthones incorporate O-heterocyclic groups, such as furan and pyran rings, into a dibenzo-γ-pirone scaffold [62]. Polycyclic xanthones are aromatic polyketide derivatives assembled from type II PKSs using malonyl-CoA as the substrate, and they have an angular hexacyclic framework that is highly oxygenated and contains a xanthone substructure and an isoquinolone or isochromane moiety [63]. The structural diversity of polycyclic xanthones depends on the variation in the oxidation state of the xanthone moiety and the diversity of substituents, including hydroxyl, halogen atoms, and sugar moieties [64]. The variation of the oxidation state of the quinone/hydroquinone in the isoquinolone moiety and the diversity of substituents, including alkyl, hydroxyl, and halogen atoms, also contribute to the structural diversity of polycyclic xanthones [64]. The methylene dioxybridge or the oxazolidine ring fused with the angular hexacyclic framework is also essential for the structural diversity of xanthones [64]. Complex xanthones include dimeric, pseudo-dimeric (one xanthonic and a hydroxanthone nucleus connected by a C-C bond), and glycosylated xanthones [64].

Albofungin (**80**) (Figure 12) is isolated from the culture medium of *Actinomyces tumemacerans* strain INMI P-42 and has been found to possess broad-spectrum antifungal activity against a range of fungal species, including *C. albicans*, *Candida guiliermondii*, *Candida hrusei*, *Candida parahrusei*, *C. tropicalis*, *Candida stellatoidea*, *C. neoformans*, *A. niger*, *Aspergillus oryzae*, and *Saccharomyces cerevisiae*, with MIC values ranging from 0.0075 to 1 μg/mL [65]. Furthermore, albofungin (**80**) has demonstrated a favorable safety profile, with a lethal dose that gives 50% mortality (LD_50_) of 2.0 mg/kg for intraperitoneal administration [65]. Sch 42137 (**81**) (Figure 12) is obtained from the culture medium of *Actinoplane* species SCC 1906 [66]. Sch 42137 (**81**) and albofungin (**80**) have a similar angular hexacyclic framework, but the methylene dioxybridge has a different fusion position. The xanthone moiety and the isoquinolin moiety of Sch 42137 (**81**) also differ from albofungin (**80**) in the oxidation state and substituents. Sch 42137 (**81**) demonstrates weaker antifungal activity against *C. albicans*, *C. tropicalis*, *C. stellatoidea*, and *C. parapsilosis* with MIC values of approximately 0.125 µg/mL [67]. Sch 54445 (**82**) (Figure 12) is obtained from the culture medium of *Actinoplane* species SCC 2314 and ATCC 55600 [67]. In contrast to albofungin (**80**), the C9-C14 double bond of the xanthone moiety of Sch 54445 (**82**) disappears, forming the C11-C12 double bond. The C-22 position of the isoquinolin moiety of Sch 54445 (**82**) connects the chlorine atom, and the 1-methylbutyl group connected at the C-25 position replaces the original methyl group. Compared to albofungin, Sch 54445 (**82**) exhibits a heightened antifungal activity against *C. albicans*, *C. tropicalis*, *C. stellatoidea*, and *C. parapsilosis* with MICs of approximately 0.00038 µg/mL. Additionally, Sch 54445 significantly demonstrates efficacy against *Aspergillus flavus*, *A. niger*, and *A. fumgatus* with MICs of 0.025 µg/mL [67]. Furthermore, Sch 54445 (**82**) has a favorable safety profile, as evidenced by an LD_50_ value of 1 mg/kg administered intravenously to mice [67]. Sch 56036 (**83**) (Figure 12) is isolated from the culture broth of an *Actinoplanes* species (SCC 2314, ATCC 55600) [68]. In contrast to albofungin (**80**), Sch56036 (**83**) only has an angular hexacyclic framework but does not fuse the methylene dioxybridge [68]. Sch56036 (**83**) exhibits antifungal activity against *C. albicans* and *C. tropicalis* (geometric mean MIC = 0.017 μg/mL) [68]. It also shows antifungal activity against *Aspergillus* (geometric mean MIC = 0.794 μg/mL) [68]. 15R-17,18-dehydroxantholipin (**84**) (Figure 12) is isolated from mangrove *Streptomyces qinglanensis* 172205 and shows antifungal activity against *C. albicans* with MIC value of 3.13 μg/mL [69] (Table 1). Turbinmicin (**85**) (Figure 12) is isolated from a sea squirt microbiome constituent, the bacterium *Micromonospora* species WMMC-415 [70]. Turbinmicin has a heptacyclic ring system like albofungin, but turbinmicin is connected to the polyene tail at the C-13 position. Turbinmicin (**85**), which hinders the transport of extracellular vesicles to the extracellular matrix and impedes the assembly of biofilm [71], targets Sec14, which is a peripheral Golgi membrane protein that is responsible for transporting phosphatidylinositol and phosphatidylcholine in cells and is crucial for lipid metabolism and membrane transportation [72].

The essentiality of the polyene side chain for the antifungal activity of turbinmicin (**85**) is demonstrated by the significant reduction in antifungal activity upon cleavage of the polyene side chain by hydrolysis [70]. Turbinmicin (**85**) exhibits noteworthy broad-spectrum antifungal activity against *C. albicans*, *C. glabrata*, *C. tropicalis*, *C. auris*, *A. fumigatus*, *Fusarium*, and *Scedosporium* species, with MIC values ranging from 0.03 to 0.5 µg/mL [70] (Table 1). Turbinmicin (**85**) significantly diminished the fungal load of the neutropenic mouse model of *C. auris* injection and the neutropenic and corticosteroid immuno-suppressed mouse model of *A. fumigatus* injection [70]. Turbinmicin (**85**), a potent antifungal lead compound, has shown significant in vivo and in vitro effectiveness, devoid of any apparent toxic effects, presenting a promising avenue for developing novel antifungal drugs.

Parnafungins A and B are isolated from the acetone extract of lichenicolous strains fermentation of *Fusarium larvarum* (Ascomycota, Hypocreales) [73]. These compounds exhibit the ability to interconvert, with the major syn relative configurations (parnafungins A1 (**86**) and B1 (**87**)) (Figure 13) and minor anti-relative configurations (parnafungins A2 (**88**) and B2 (**89**)) (Figure 13) being determined based on the relative configuration of C15-hydroxyl and C15A-methyl carboxylate [73]. The biological activity of the parnafungins A and B mixture is dependent on the presence of the intact isoxazolidinone ring. Despite the broad-spectrum antifungal activity exhibited by the mixture of parnafungins A and B against various *Candida* species, their efficacy is limited due to the inherent instability of the isoxazolidinone ring, resulting in the loss of antifungal activity upon ring-opening (**90** and **91**) (Figure 13) [73]. Through the utilization of affinity selection/mass spectrometry, it has been discovered that the “straight” structural isomer of parnafungin A1 (**86**) exhibits a higher affinity for polyadenosine polymerase compared to the “bent” structural isomer of parnafungin B1 (**87**) [74]. The *C. albicans* fitness test demonstrated that a parnafungin A and B mixture inhibits polyadenosine polymerase, a key component of the fungal mRNA cleavage and polyadenylation complex. In a mouse model of disseminated *candidiasis*, treatment with parnafungins at a 50 mg/kg dosage reduced renal fungal burden and showed in vivo efficacy without any observable toxicity [75]. Parnafungins C (**92**) and D (**93**) (Figure 13), analogs of parnafungin A, are isolated from the acetone extract of *Fusarium larvarum* strain F-155, 597. Parnafungin C (**92**) is produced through the methylation of the C7-phenolic hydroxyl group of parnafungin A, and parnafungin D (**93**) is obtained by methylation of the C7-phenolic hydroxyl group and addition of an epoxide to the xanthone structure [76]. Parnafungin C (**92**) exhibits antifungal activity against *C. albicans*, *C. lusitaniae*, *C. krusei*, and *C. tropicalis* with MIC values ranging from 0.08 to 2.5 μg/mL, while parnafungin D (**93**) shows antifungal activity against *C. albicans*, *C. glabrata*, *C. parapsilosis*, *C. lusitaniae*, *C. krusei*, and *C. tropicalis* with MIC values ranging from 0.016 to 5 μg/mL [76] (Table 1).

## 5. Linear Polyketides

The polyketide chain skeleton was catalyzed by type I PKSs and then subjected to complex post-modifications, including glycosylation, to form linear polyketide [77]. There are differences in the sphingolipid synthesis pathways between fungi and mammals. Fungi utilize a process whereby phosphoinositide is transferred to the 1-OH group of ceramides to generate inositol phosphoceramide rather than directly producing sphingesters [78]. Khafrefungin (**94**) (Figure 14), isolated from endophytic fungi found in a Costa Rican plant, has been found to inhibit the inositol phosphoceramide synthase of *S. cerevisiae* and pathogenic fungi. This inhibition leads to the blockage of the phosphoinositide-to-ceramide pathway, thereby inhibiting fungal sphingolipid synthesis while leaving t mammalian sphingolipid synthesis unaffected [78]. Khafrefungin (**94**) shows antifungal activity against *C. albicans*, *C. neoformans*, and *S. cerevisiae*, with MIC values of 2, 2, and 15.6 μg/mL, respectively [78]. Additionally, khafrefungin (**94**) has been shown to possess fungicidal activity against *C. albicans*, *C. neoformans*, and *S. cerevisiae* with minimum fungicidal concentrations of 4, 4, and 15.6 μg/mL, respectively [78]. Notably, the removal of the aldonic acid group (**95**) (Figure 14) greatly attenuates its antifungal activity against *S. cerevisiae* (MIC > 200 µM), indicating the importance of the aldonic acid group for the antifungal activity of khafrefungin (**94**) [79]. The presence of the enantiomeric form of the aldonic acid group (**96**) (Figure 14) has been found to diminish the antifungal activity of khafrefungin (**94**), indicating that the aldonic acid group not only enhances the water solubility of khafrefungin (**94**) but also plays a role in its antifungal activity [79]. Furthermore, the enantiomer of the 4-methyl group (**97**) (Figure 14) has been shown to completely abolish the activity of khafrefungin (**94**), highlighting the essentiality of the configuration of the 4-methyl group for the antifungal activity of khafrefungin (**94**) [79]. Additionally, treating khafrefungin (**94**) under acidic conditions forms a six-membered lactone derivative (**98**) (Figure 14) that exhibits comparable antifungal activity against *S. cerevisiae* (MIC = ~10 µM) to that of native khafrefungin (**94**) [80] (Table 1).

Basiliskamide A and B are isolated from *Bacillus laterosporus* PNG 276 [81]. Basiliskamide A (**99**) (Figure 15) and amphotericin B (**62**) were administered to human diploid fibroblasts, with basiliskamide A (**99**) exhibiting minimal cytotoxicity at a concentration of 100 μg/mL, while cytopathic effect was observed with amphotericin B (**62**) only 12.5 μg/mL [81]. Furthermore, basiliskamide A (**99**) demonstrated antifungal activity against *C. albicans* and *A. fumigatus*, with MIC values of 1 and 2.5 μg/mL, respectively, whereas basiliskamide B (**100**) (Figure 15) exhibited antifungal activity against the same fungal strains with MIC values of 3.1 and 5 μg/mL, respectively [81]. YM-45722 (**101**) (Figure 15), an analog of basiliskamide A (**99**), inhibits the growth of *C. albicans* at a concentration of 25 μg/mL and has no effect on the growth of *A. fumigatus* at a concentration of 50 μg/mL [81] (Table 1). The linear polyketide chain of YM-45722 (**101**) exhibits an additional methylene group compared to the linear polyketide chain of basiliskamide A (**99**), potentially accounting for the diminished antifungal activity of YM-45722 (**101**) [81].

## 6. Hybrid Polyketide Nonribosomal Peptides

Echinocandins are novel lipopeptide antifungal products synthesized by a heterozygous pathway of non-ribosomal peptides synthases (NRPSs)-PKSs. Compared with azole and polyene antibiotics, echinocandins have a completely different mechanism of action and exert their antifungal effect by destroying the cell wall. Non-competitive binding of echinocandins to the catalytic subunits of β-(1,3)-D-glucan synthetase encoded by the *FKS1* and *FKS2* genes results in the inhibition of biosynthesis of β-(1,3)-D-glucan, an important component of the fungal cell wall, which destroys the integrity of fungal cell wall and disrupts the osmotic balance, ultimately leading to fungal death [82,83]. Echinocandins show great in vitro antifungal activity against various invasive fungal pathogens, including *Candida* and *Aspergillus* species, but they are ineffective against *C. neoformans* [84] (Table 1). Currently, four echinocandin antifungal drugs are on the market, including caspofungin, micafungin, anidulafungin, and rezafungin. Echinocandin B, the lead compound of Anidulafungin and rezafungin, and FR901379, the lead compound of micafungin, are assembled through the NRPs and fatty acid synthases heterozygous pathway [85,86]. Only pneumocandin B_0_ (**102**) (Figure 16), the lead compound of caspofungin (**103**) (Figure 16), is assembled by the NRPSs and PKSs heterozygous pathway [85]. Only caspofungin and its lead compound pneumocandin B_0_ (**102**) are discussed in this section. Pneumocandin B_0_ (**102**) is isolated from the filamentous fungus *Glarea lozoyensis* [87,88]. Pneumocandin B_0_ (**102**) is a lipopeptide composed of myristic acid and a hexapeptide ring. PKSs catalyze the assembly of 10, 12-dimethylmyristic acid, and then, catalyzed by a series of enzymes, the polyketide intermediate localizes to NRPSs to acylate the 4, 5-dihydroxyornithine of pneumocandin B_0_ (**102**), initiating cyclic hexapeptide elongation.

Burkholdines 1229 (**104**) and 1097 (**105**) (Figure 17) are isolated from the bacteria *Burkholderia ambifaria* 2.2N [89]. Burkholdine 1229 exhibits potent antifungal activity against *S. cerevisiae* (MIC = 0.4 μg/mL), *C. albicans* (MIC = 12.5 μg/mL), and *A. niger* (MIC = 12.5 μg/mL) [89]. Burkholdine 1097 exhibits potent antifungal activity against *S. cerevisiae* (MIC = 1.6 μg/mL), *A. niger* (MIC = 1.6 μg/mL), and *C. albicans* (MIC = 12.5 μg/mL). Burkholdine 1215 (**106**), 1119 (**107**), and 1213 (**108**) (Figure 17) are isolated from the bacteria *Burkholderia ambifaria* [90]. Burkholdine 1215 (**106**) exhibits potent antifungal activity against *S. cerevisiae* (MIC = 0.15 μg/mL), *C. albicans* (MIC = 0.15 μg/mL), and *A. niger* (MIC = 0.15 μg/mL) [90]. Burkholdine 1119 (**107**) exhibits potent antifungal activity against *S. cerevisiae* (MIC = 0.1 μg/mL), *C. albicans* (MIC = 0.4 μg/mL), and *A. niger* (MIC = 0.1 μg/mL) [90]. Burkholdine 1213 (**108**) exhibits potent antifungal activity against *S. cerevisiae* (MIC = 2.0 μg/mL), *C. albicans* (MIC = 31.0 μg/mL), and *A. niger* (MIC = 2.0 μg/mL) [90]. The antifungal activity of burkholdines 1215 (**106**) and 1119 (**107**) containing 2,4-diaminobutyric acid adjacent to the 3-OH-Tyr structure is higher than that of burkholdines 1229 (**104**), 1097 (**105**), and 1213 (**108**) containing Asn adjacent to the 3-OH-Tyr structure [90]. The antifungal activity of burkholdines 1215 (**106**) and 1119 (**107**) is significantly higher than hemolytic activity, but the antifungal activity of burkholdines 1229 (**104**), 1097 (**105**), and 1213 (**108**) is equivalent to hemolytic activity [90]. Burkholdine 1119 (**107**), assigning an Asn replaced a 3-OH-Asn and attaching a β-xyloside glycosyl portion replaced a α-xyloside glycosyl portion compared to burkholdine 1215 (**106**), shows weaker activity against *C. albicans* (MIC = 0.4 μg/mL) and stronger activity against *A. niger* (MIC = 0.1 μg/mL). Burkholdine 1213 (**108**), attaching an Asn replaced a 2,4-aminobutyric acid of burkholdine 1119 (**107**), shows weaker activity against both *C. albicans* (MIC = 31.0 μg/mL) and *A. niger* (MIC = 2.0 μg/mL). Burkholdine 1229 (**104**), assigning a 3-OH-Asn replaced an Asn of burkholdine 1213(**108**), shows stronger activity against *C. albicans* (MIC = 12.5 μg/mL) and weaker activity against *A. niger* (MIC = 12.5 μg/mL). Burkholdine 1097 (**105**), removing the β-xyloside glycosyl portion of burkholdine 1229 (**104**), shows the same activity against *C. albicans* (MIC = 12.5 μg/mL) and stronger activity against *A. niger* (MIC = 1.6 μg/mL) (Table 1).

## 7. Pyridine Derivatives

Funiculosin (**109**) (Figure 18), a neutral lipophilic compound isolated from the fermentation of *Penicillium funiculosum* T_HOM_ [91], has been found to bind to cytochrome b-asparagine-208 strongly and inhibit the activity of mitochondrial bc1 complex, ultimately leading to cell death [92,93] and with a broad spectrum of antifungal activity, including *C. albicans* (MIC = 2 µg/mL), *C. utilis* (MIC = 5 µg/mL), *C. neoformans* (MIC = 2 µg/mL), *S. cerevisiae* (MIC = 20 µg/mL), *T. mentagrophytes* (MIC = 15 µg/mL), *T. rubrun* (MIC = 3.9 µg/mL), *M. gypseum* (MIC < 0.12 µg/mL), *Blastomyces derniatitidis* (MIC < 0.12 µg/mL), and *Hormodendrum pedrosoi* (MIC = 15 µg/mL) [91,94] (Table 1). Funiculosin (**109**) has demonstrated significant potent efficacy in vivo, as evidenced by its ability to effectively treat *T. mentagrophytes* hyphae infection in guinea pigs. Specifically, a hydrophilic ointment containing 0.5% funiculosin had a cure rate of 97% within 10 days [94]. Notably, the toxicity of funiculosin (**109**) to guinea pigs and rabbits was negligible, as evidenced by the survival of guinea pigs injected intraperitoneally with 500 mg/kg of the compound. However, it should be noted that funiculosin is toxic to mice and rats, with an LD_50_ of 5–7 mg/kg for both oral and intraperitoneal administration [94].

## 8. Other Polyketides

Emodin (**110**) and skyrin (**111**) (Figure 19) are anthraquinones isolated from the mangrove endophytic fungus *Talaromyces* species ZH-154 [95]. Emodin (**110**) inhibits the growth of *C. albicans* ATCC 10231 (MIC = 6.25 μg/mL), *A. niger* ATCC 13496 (MIC = 12.5 μg/mL), and skyrin (**111**) inhibits the growth of *C. albicans* ATCC 10231 (MIC = 12.5 μg/mL) [95]. Hippolachnin A (**112**) (Figure 19) is a furan derivative isolated from the South China Sea sponge *Hippospongia lachne* and exhibits potent antifungal activity against *C. neoformans* with a MIC of 0.41 μM [96]. Wortmannin (**113**) (Figure 19) is a furan derivative isolated from the ethyl acetate extract of *Fusarium oxysporum* (N17B) and exhibits potent antifungal activity against *C. albicans* with a MIC of 0.78 μg/mL [97]. Wortmannin (**113**) inhibits mammalian phosphatidylinositol 3-kinase in vitro and in vivo and membrane-associated phosphatidylinositol 4-kinase of the fungus *Schizosacchromyces pombe* [98]. Microketides A (**114**) and B (**115**) (Figure 19) are isolated from the gorgonian-derived fungus *Microsphaeropsis* species RA10-14 collected from the South China Sea [99]. Microketides A (**114**) and B (**115**) have great broad-spectrum antifungal activity against *C. albicans*, *Colletotrichum truncatum*, *Gloeosporium musarum*, and *Pestalotia calabae* (MICs = 1.56–3.13 μg/mL) [99]. Macrotermycins A (**116**) and C (**117**) (Figure 19) are macrolactam polyketides isolated from a termite-associated actinomycete, *Amycolatopsis* species M39 [100]. Macrotermycin A (**116**) shows antifungal activity against *C. albicans* ATCC 10231 (MIC = 10 μg/mL) and *S. cerevisiae* ATCC9763 (MIC = 5 μg/mL) [100]. Macrotermycin C (**117**) shows weaker antifungal activity against *C. albicans* ATCC 10231 (MIC = 25 μg/mL) and *S. cerevisiae* ATCC9763 (MIC = 20 μg/mL) [100]. F2928-1 (**118**), hakuhybotrol (**119**), cladobotric acid A (**120**), cladobotric acid F (**121**), cladobotric acid E (**122**), cladobotric acid H (**123**), and pyrenulic acid A (**124**) (Figure 19) are isolated from the cultured material of the mycoparasitic fungus *Hypomyces pseudocorticiicola* FKA-73 [101]. These compounds show potent antifungal activity against azole-sensitive and azole-resistant *C. auris*, *A. fumigatus*, *A. udagawae*, *A. felis*, and *A. lentulus* [101]. Campafungins A (**125**), B (**126**), C (**127**), and D (**128**) (Figure 19) are isolated from the *Plenodomus enteroleucus* Strain F-146,176. The compounds exhibit moderate antifungal activity against *C. neoformans*, with MIC values ranging from 4 to 8 μg/mL [102]. Campafungins A (**125**), B (**126**), C (**127**), and D (**128**) show antifungal activity against *C. neoformans* H99 with MIC values of 8 μg/mL, 4 μg/mL, 4 μg/mL, and 8 μg/mL, respectively (Table 1).

**Table 1 biomolecules-13-01572-t001:** The general characteristic of poliketides having antifungal activity.

Compound Class	No.	Compound Name	Source	Target	References
Macrolide polyketides	1	Amphidinin Q	dinoflagellates *Amphidinium* species (2012-7-4A strain)	*C. albicans* MIC = 32 μg/mL	[19]
	2	Amphidinin C	dinoflagellates *Amphidinium* species (2012-7-4A strain)	*A. niger* MIC = 32 μg/mL	[19]
	3	Amphidinin E	dinoflagellates *Amphidinium* species (2012-7-4A strain)	*A. niger* MIC =16 μg/mL	[19]
	4	Rustmicin	*Micromonospora narashinoensis* 980-MC	*C. neoformans* MY2062 MIC = 0.0001 μg/mL *C. neoformans* ATCC9011 MIC = 0.063 μg/mL *C. tropicalis* MY1012 MIC = 0.05 μg/mL *C. albicans* MY1055 MIC = 6.25 μg/mL *C. albicans* ATCC90028 MIC = 4 μg/mL	[22,23]
	5	-	-	*C. neoformans* ATCC9011 MIC = 0.5 μg/mL *C. albicans* ATCC90028 MIC = 64 μg/mL	[23]
	6	-	-	*C. neoformans* ATCC9011 MIC > 64 μg/mL *C. albicans* ATCC90028 MIC > 64 μg/mL	[23]
	7	-	-	*C. neoformans* ATCC9011 MIC = 32 μg/mL *C. albicans* ATCC90028 MIC > 64 μg/mL	[23]
	8	-	-	*C. neoformans* ATCC9011 MIC = 16 μg/mL *C. albicans* ATCC90028 MIC > 64 μg/mL	[23]
	9	-	-	*C. neoformans* ATCC9011 MIC > 64 μg/mL *C. albicans* ATCC90028 MIC > 64 μg/mL	[23]
	10	-	-	*C. neoformans* ATCC9011 MIC > 64 μg/mL *C. albicans* ATCC90028 MIC > 64 μg/mL	[23]
	11	21-Hydroxy rustmicin	*Micromonospora* species UV Mutant (MA 7186)	*C. tropicalis* MY1012 MIC = 0.024 μg/mL *C. albicans* MY1055 MIC = 12.5 μg/mL *C. neoformans* MY2062 MIC = 0.1 μg/mL	[22]
	12	Galbonolide B	*Micromonospora* species (MA 7094) and UV Mutant (MA 7186)	*C. neoformans* MY2062 MIC = 12.5 μg/mL *C. tropicalis* MY1012 MIC = 200 μg/mL *C. albicans* MY1055 MIC > 200 μg/mL	[22]
	13	21-Hydroxy galbonolide B	*Micromonospora* species UV Mutant (MA 7186)	*C. tropicalis* MY1012 MIC = 0.78 μg/mL *C. neoformans* MY2062 MIC = 3.1 μg/mL	[22]
	14	Preussolide A	*Preussia typharum*	*C. neoformans* H99 (37 °C) MIC = 4 μg/mL *C. neoformans* H99 (23 °C) MIC = 8 μg/mL *C. albicans* ATCC 10231 MIC = 256 μg/mL *A. fumigatus* AF239 MIC = 8 μg/mL	[24]
	15	Preussolide B	*Preussia typharum*	*C. neoformans* H99 (37 °C) MIC = 32 μg/mL *C. neoformans* H99 (23 °C) MIC = 32 μg/mL *C. albicans* ATCC 10231 MIC = 256 μg/mL	[24]
	16	Oligomycin A *s-Trans*	*Streptomyces*	*C. albicans* ATCC 24433 MIC = 2–4 μg/mL *C. krusei* 432M (MIC = 1–2 μg/mL) *C. parapsilosis* ATCC 22019 MIC = 2 μg/mL *C. utilis* 84 MIC = 1 μg/mL *C. tropicalis* 3019 MIC = 1 μg/mL *A. niger* MIC = 0.5–2 μg/mL *C. humicolus* ATCC 9949 MIC = 2 μg/mL *T. mentagrophytes* ATCC 9533 MIC = 10 μg/mL *C. albicans* ATCC 14053 MIC = 4 μg/mL *A. niger* ATCC 16404 MIC = 0.125 μg/mL *C. albicans* ATCC 14053 MIC = 4 μg/mL *C. humicolus* ATCC 9949 MIC = 2 μg/mL *A. niger* ATCC 10335 MIC = 2 μg/mL *A. niger* MIC = 0.125 μg/mL	[25,26,27,28,29,30,31,33]
	17	Oligomycin A *s-Cis*	*Streptomyces*	*C. albicans* ATCC 24433 MIC = 2–4 μg/mL *C. krusei* 432M MIC = 1–2 μg/mL *C. parapsilosis* ATCC 22019 MIC = 2 μg/mL *C. utilis* 84 MIC = 1 μg/mL *C. tropicalis* 3019 MIC = 1 μg/mL *A. niger* MIC = 0.5–2 μg/mL *C. humicolus* ATCC 9949 (MIC = 2 μg/mL) *Aspergillus carbonarius* M333 (MIC = 2 μg/mL) *A. westerdijkiae* NRRL 3174 (MIC = 8 μg/mL) *A. parasiticus* CBS 100926 (MIC = 4 μg/mL) *A. nidulans* KE202 (MIC = 75 μg/mL) *A. niger* OT304 (MIC = 4 μg/mL) *A. terreus* CT290 (MIC = 75 μg/mL) *A. fumigatus* CF140 (MIC = 100 μg/mL)	[25,26,27,28,29,30,32]
	18	-	-	*C. albicans* ATCC 24433 (MIC > 32 μg/mL) *C. parapsilosis* ATCC 22019 (MIC > 32 μg/mL) *C. krusei* 432M (MIC > 32 μg/mL) *A. niger* 137a (MIC > 32 μg/mL)	[26]
	19	-	-	*C. albicans* ATCC 24433 (MIC > 32 μg/mL) *C. parapsilosis* ATCC 22019 (MIC > 32 μg/mL) *C. krusei* 432M (MIC > 32 μg/mL) *A. niger* 137a (MIC > 32 μg/mL)	[26]
	20	-	-	*C. albicans* ATCC 24433 (MIC > 32 μg/mL) *C. parapsilosis* ATCC 22019 (MIC > 32 μg/mL) *C. krusei* 432M (MIC > 32 μg/mL) *A. niger* 137a (MIC > 32 μg/mL)	[26]
	21	(33S)-oligomycin A	-	*C. parapsilosis* ATCC 22019 MIC = 1 μg/mL *C. albicans* ATCC 24433 MIC = 4 μg/mL *C. utilis* 84 MIC = 2 μg/mL *C. tropicalis* 3019 MIC = 1 μg/mL *C. krusei* 432 M MIC = 4 μg/mL *A. niger* 137 a MIC = 2 μg/mL	[28]
	22	-	-	*C. albicans* ATCC 14053 MIC = 16 μg/mL *A. niger* ATCC 16404 MIC = 4 μg/mL	[31]
	23	Bromo- oligomycin A	-	*A. niger* ATCC 16404 MIC >16 μg/mL *C. albicans* ATCC 14053 MIC > 16 μg/mL *C. humicolus* MIC = 2 μg/mL	[29]
	24	Oligomycin E	*Streptomyces* species strain HG29	*Aspergillus carbonarius* M333 (MIC = 2 μg/mL) *A. westerdijkiae* NRRL 3174 (MIC = 8 μg/mL) *A. parasiticus* CBS 100926 (MIC = 4 μg/mL) *A. nidulans* KE202 (MIC = 75 μg/mL) *A. niger* OT304 (MIC = 4 μg/mL) *A. terreus* CT290 (MIC = 75 μg/mL) *A. fumigatus* CF140 (MIC = 100 μg/mL)	[32]
	25	Oligomycin C	*Streptomyces diastaticus*	*A. niger* ATCC 10335 MIC = 2 μg/mL	[30]
	26	-	-	-	[33]
	27	-	-	*A. niger* MIC = 2 μg/mL	[33]
	28	-	-	*A. niger* MIC = 2 μg/mL	[33]
	29	Neomaclafungin A	*Actinoalloteichus* species NPS702	*T. mentagrophytes* ATCC 9533 MIC = 3 μg/mL	[27]
	30	Neomaclafungin B	*Actinoalloteichus* species NPS702	*T. mentagrophytes* ATCC 9533 MIC = 3 μg/mL	[27]
	31	Neomaclafungin C	*Actinoalloteichus* species NPS702	*T. mentagrophytes* ATCC 9533 MIC = 1 μg/mL	[27]
	32	Neomaclafungin D	*Actinoalloteichus* species NPS702	*T. mentagrophytes* ATCC 9533 MIC = 1 μg/mL	[27]
	33	Neomaclafungin E	*Actinoalloteichus* species NPS702	*T. mentagrophytes* ATCC 9533 MIC = 1 μg/mL	[27]
	34	Neomaclafungin F	*Actinoalloteichus* species NPS702	*T. mentagrophytes* ATCC 9533 MIC = 3 μg/mL	[27]
	35	Neomaclafungin G	*Actinoalloteichus* species NPS702	*T. mentagrophytes* ATCC 9533 MIC = 3 μg/mL	[27]
	36	Neomaclafungin H	*Actinoalloteichus* species NPS702	*T. mentagrophytes* ATCC 9533 MIC = 3 μg/mL	[27]
	37	Neomaclafungin I	*Actinoalloteichus* species NPS702	*T. mentagrophytes* ATCC 9533 MIC = 3 μg/mL	[27]
	38	Brasilinolide A	*Nocardia brasiliensis* IFM0406	*A. niger* IFM 40406 MIC = 3.13 μg/mL	[34]
	39	Brasilinolide B	*Nocardia brasiliensis* IFM0406	*A. niger* MIC = 12.5 μg/mL *A. fumigatus* IFM 41219 MIC = 12.5 μg/mL *C. albicans* ATCC 90028 MIC = 25 μg/mL *C. albicans* IFM 40007 MIC = 12.5 μg/mL *C. albicans* 94–2530 MIC = 25 μg/mL *C. krusei* M 1005 MIC = 25 μg/mL *C. parapsilosis* ATCC 90018 MIC = 12.5 μg/mL *C. glabrata* ATCC 90030 MIC = 25 μg/mL *C. neoformans* ATCC 90112 MIC = 12.5 μg/mL *C. neoformans* 145 A MIC = 25 μg/mL	[35]
	40	Copiamycin		*C. albicans* Yu 1200 MIC = 25 μg/mL	[36]
	41	Methylcopiamycin	-	*C. albicans* Yu 1200 MIC = 25 μg/mL	[36]
	42	Demalonylmethylcopiamycin	-	*C. albicans* Yu 1200 MIC = 6.25 μg/mL	[36]
	43	Langkolide	*Streptomyces* species Acta 3062	*Candida glabrata* IC_50_ = 1.00 ± 0.02 μM *C. albicans* IC_50_ = 1.23 ± 0.10 μM	[37]
	44	Cyphomycin	*Brazilian Streptomyces* ISID311	*A. fumigatus* 11628 MIC = 0.5 μg/mL *C. glabrata* 4720 MIC = 0.5 μg/mL *C. auris* B11211 MIC = 4 μg/mL	[38]
	45	Caniferolide A	*Streptomyces caniferus* CA-271066	*A. fumigatus* ATCC46645 MIC = 2–4 μg/mL *C. albicans* MY1005 MIC = 0.5–1 μg/mL	[39]
	46	Caniferolide B	*Streptomyces caniferus* CA-271066	*A. fumigatus* ATCC46645 MIC = 2–4 μg/mL *C. albicans* MY1005 MIC = 1–2 μg/mL	[39]
	47	Caniferolide C	*Streptomyces caniferus* CA-271066	*A. fumigatus* ATCC46645 MIC = 4–8 μg/mL *C. albicans* MY1005 MIC = 0.5–1 μg/mL	[39]
	48	Caniferolide D	*Streptomyces caniferus* CA-271066	*A. fumigatus* ATCC46645 MIC = 4–8 μg/mL *C. albicans* MY1005 MIC = 0.5–1 μg/mL	[39]
	49	Iseolide A	*Streptomyces* species DC4-5	*C. albicans* NBRC0197 MIC = 0.39 μg/mL	[40]
	50	Iseolide B	*Streptomyces* species DC4-5	*C. albicans* NBRC0197 MIC = 6.25 μg/mL	[40]
	51	Iseolide C	*Streptomyces* species DC4-5	*C. albicans* NBRC0197 MIC = 3.16 μg/mL	[40]
	52	Astolide A	*Streptomyces hygroscopicus*	*C. albicans* ATCC 14053 (MIC = 2.5 μg/mL) *A. niger* ATCC 16404 (MIC = 1.25 μg/mL) *C. albicans* 1582 (MIC = 2.53 μg/mL) *C. tropicales* 1402 (MIC = 5.06 μg/mL) *A. niger* 219 (MIC = 2.53 μg/mL)	[41]
	53	Astolide B	*Streptomyces hygroscopicus*	*C. albicans* ATCC 14053 (MIC = 1.25 μg/mL) *A. niger* ATCC 16404 (MIC = 0.6 μg/mL) *C. albicans* 1582 (MIC = 2.51 μg/mL) *C. tropicales* 1402 (MIC = 5.01 μg/mL) *A. niger* 219 (MIC = 2.51 μg/mL)	[41]
	54	Guanidylfungin A	*Streptomyces hygroscopicus* No. 662	*C. albicans* IAM 4888 MIC = 12.5 μg/mL *C. albicans* Yu 1200 MIC = 50 μg/mL *A. fumigatus* IAM 2153 MIC = 25 μg/mL	[36,42]
	55	Methylguanidylfungin A	-	*C. albicans* IAM 4888 MIC = 25 μg/mL *C. albicans* Yu 1200 MIC = 50 μg/mL *A. fumigatus* IAM 2153 MIC = 12.5 μg/mL	[36]
	56	Ethyl- guanidylfungin A	-	*C. albicans* IAM 4888 MIC = 25 μg/mL *A. fumigatus* IAM 2153 MIC = 25 μg/mL	[36]
	57	Butyl- guanidylfungin A	-	*C. albicans* IAM 4888 MIC = 25 μg/mL *A. fumigatus* IAM 2153 MIC = 50 μg/mL	[36]
	58	Allyl- guanidylfungin A	-	*C. albicans* IAM 4888 MIC = 25 μg/mL *A. fumigatus* IAM 2153 MIC = 25 μg/mL	[36]
	59	-	-	*C. albicans* IAM 4888 MIC = 3.12 μg/mL *C. albicans* Yu 1200 MIC = 6.25 μg/mL *A. fumigatus* IAM 2153 MIC = 3.12 μg/mL	[36]
	60	-	-	*C. albicans* IAM 4888 MIC > 100 μg/mL *C. albicans* Yu 1200 MIC > 100 μg/mL *A. fumigatus* IAM 2153 MIC = 50 μg/mL	[36]
	61	-	-	*C. albicans* IAM 4888 MIC = 100 μg/mL *C. albicans* Yu 1200 MIC = 100 μg/mL *A. fumigatus* IAM 2153 MIC = 12.5 μg/mL	[36]
	62	Amphotericin B	*Streptomyces nodosus*	*C. neoformans*, *Candida* species, and *A. fumigatus*	[7,8]
Polyether polyketides	63	Amphidinol 3	dinoflagellate *Amphidinium klebsii*	*A. niger* MEC = 8 μg/disk	[52,53]
	64	-	-	*A. niger* MEC = 20 μg/disk	[52]
	65	-	-	-	[52]
	66	Amphidinol 18	Dinoflagellate *Amphidinium carterae*	*C. albicans* MIC = 9 μg/mL	[54]
	67	Amphidinol A	dinoflagellate *Amphidinium carterae*	*C. albicans* MIC = 19 μg/mL	[51]
	68	Karatungiol A	marine dinoflagellates	*A. niger*	[55]
	69	Amphidinol 6	dinoflagellate *Amphidinium klebsii*	*A. niger* MEC = 6 μg/disk	[53]
	70	Amphidinol 2	dinoflagellate *Amphidinium klebsii*	*A. niger* MEC = 6 μg/disk	[53]
	71	Amphidinol 7	dinoflagellate *Amphidinium klebsii*	*A. niger* MEC = 10 μg/disk	[53]
	72	Desulfurization amphidinol 7	-	*A. niger* MEC = 8 μg/disk	[53]
	73	Amphidinol 20	dinoflagellate *Amphidinium carterae*	*A. niger*	[56]
	74	Amphidinol 21	dinoflagellate *Amphidinium carterae*	*A. niger*	[56]
	75	Carteraol E	marine dinoflagellates	*A. niger* MEC = 15 μg/disk	[57]
	76	Yessotoxin	dinoflagellate *Protoceratium reticulatum*	*A. niger*	[58]
	77	Desulfated yessotoxin	-	*A. niger*	[58]
	78	Hydrogen-desufated yessotoxin	-	*A. niger*	[58]
	79	Forazoline A	*Actinomadura* species strain WMMB-499	*C. albicans* K1 MIC = 16 μg/mL	[60].
Xanthone polyketides	80	Albofungin	*Actinomyces tumemacerans* strain INMI P-42	*C. albicans*, *Candida guiliermondii*, *Candida hrusei*, *Candida parahrusei*, *C. tropicalis*, *Candida stellatoidea*, *C. neoformans*, *A. niger*, *Aspergillus oryzae*, and *Saccharomyces cerevisiae* MICs = 0.0075–1 μg/mL	[65]
	81	Sch 42137	*Actinoplane* species SCC 1906	*C. albicans*, *C. tropicalis*, *C. stellatoidea*, and *C. parapsilosis* MICs = 0.125 µg/mL	[66]
	82	Sch 54445	*Actinoplane* species SCC 2314 and ATCC 55600	*C. albicans*, *C. tropicalis*, *C. stellatoidea*, and *C. parapsilosis* MICs = 0.00038 µg/mL	[67]
	83	Sch 56036	*Actinoplanes* species (SCC 2314, ATCC 55600)	*C. albicans* and *C. tropicalis* MICs = 0.017 μg/mL	[68]
	84	15R-17,18-dehydroxantholipin	mangrove *Streptomyces qinglanensis* 172205	*C. albicans* MIC = 3.13 μg/mL	[69]
	85	Turbinmicin	bacterium *Micromonospora* species WMMC-415	*C. albicans*, *C. glabrata*, *C. tropicalis*, *C. auris*, *A. fumigatus*, *Fusarium* and *Scedosporium* species MICs = 0.03–0.5 µg/mL	[70]
	86	Parnafungin A1	*Fusarium larvarum*	various *Candida* species	[73]
	87	Parnafungin B1	*Fusarium larvarum*	various *Candida* species	[73]
	88	Parnafungin A2	*Fusarium larvarum*	various *Candida* species	[73]
	89	Parnafungin B2	*Fusarium larvarum*	various *Candida* species	[73]
	90	-	-	-	[73]
	91	-	-	-	[73]
	92	Parnafungin C	*Fusarium larvarum* strain F-155,597	*C. albicans*, *C. lusitaniae*, *C. krusei*, and *C. tropicalis* MICs = 0.08–2.5 μg/mL	[76]
	93	Parnafungin D	*Fusarium larvarum* strain F-155,597	*C. albicans*, *C. glabrata*, *C. parapsilosis*, *C. lusitaniae*, *C. krusei*, and *C. tropicalis* MICs = 0.016–5 μg/mL	[76]
Linear polyketides	94	Khafrefungin	endophytic fungi	*C. albicans* MIC = 2 μg/mL *C. neoformans* MIC = 2 μg/mL *S. cerevisiae* MIC = 15.6 μg/mL	[78]
	95	-	-	*S. cerevisiae* MIC > 200 µM	[79]
	96	-	-	-	[79]
	97	-	-	-	[79]
	98	-	-	*S. cerevisiae* MIC = ~10 µM	[80]
	99	Basiliskamide A	*Bacillus laterosporus* PNG 276	*C. albicans* MIC = 1 μg/mL *A. fumigatus* MIC = 2.5 μg/mL	[81]
	100	Basiliskamide B	*Bacillus laterosporus* PNG 276	*C. albicans* MIC = 3.1 μg/mL *A. fumigatus* MIC = 5 μg/mL	[81]
	101	YM-45722	-	*C. albicans* MIC = 25 μg/mL *A. fumigatus* MIC > 50 μg/mL	[81]
Hybrid polyketide nonribosomal peptides	102	Pneumocandin B_0_	fungus *Glarea lozoyensis*	*Candida* and *Aspergillus species*	[87,88]
	103	caspofungin	-	*Candida* and *Aspergillus species*	[84]
	104	Burkholdine 1229	bacteria *Burkholderia ambifaria* 2.2N	*S. cerevisiae* MIC = 0.4 μg/mL *C. albicans* MIC = 12.5 μg/mL *A. niger* MIC = 12.5 μg/mL	[89]
	105	Burkholdine 1097	bacteria *Burkholderia ambifaria* 2.2N	*S. cerevisiae* MIC = 1.6 μg/mL *A. niger* MIC = 1.6 μg/mL *C. albicans* MIC = 12.5 μg/mL	[89]
	106	Burkholdine 1215	bacteria *Burkholderia ambifaria*	*S. cerevisiae* MIC = 0.15 μg/mL *C. albicans* MIC = 0.15 μg/mL *A. niger* MIC = 0.15 μg/mL	[90]
	107	Burkholdine 1119	bacteria *Burkholderia ambifaria*	*S. cerevisiae* MIC = 0.1 μg/mL *C. albicans* MIC = 0.4 μg/mL *A. niger* MIC = 0.1 μg/mL	[90]
	108	Burkholdine 1213	bacteria *Burkholderia ambifaria*	*S. cerevisiae* MIC = 2.0 μg/mL *C. albicans* MIC = 31.0 μg/mL *A. niger* MIC = 2.0 μg/mL	[90]
Pyridine derivatives	109	Funiculosin	*Penicillium funiculosum* T_HOM_	*C. albicans* (MIC = 2 µg/mL) *C. utilis* (MIC = 5 µg/mL) *C. neoformans* (MIC = 2 µg/mL) *S. cerevisiae* (MIC = 20 µg/mL) *T. mentagrophytes* (MIC = 15 µg/mL) *T. rubrun* (MIC = 3.9 µg/mL) *M. gypseum* (MIC < 0.12 µg/mL) *B. derniatitidis* (MIC < 0.12 µg/mL) *H. pedrosoi* (MIC = 15 µg/mL)	[91,94]
Other polyketides	110	Emodin	fungus *Talaromyces* species ZH-154	*C. albicans* ATCC 10231 MIC = 6.25 μg/mL *A. niger* ATCC 13496 MIC = 12.5 μg/mL	[95]
	111	Dkyrin	fungus *Talaromyces* species ZH-154	*C. albicans* ATCC 10231 MIC = 12.5 μg/mL	[95]
	112	Hippolachnin A	sponge *Hippospongia lachne*	*C. neoformans* MIC = 0.41 μM	[96]
	113	Wortmannin	*Fusarium oxysporum* N17B	*C. albicans* MIC = 0.78 μg/mL	[97]
	114	Microketide A	fungus *Microsphaeropsis* species RA10-14	*C. albicans*, *C. truncatum*, *G. musarum*, and *P. calabae* MICs = 1.56–3.13 μg/mL	[99]
	115	Microketide B	fungus *Microsphaeropsis* species RA10-14	*C. albicans*, *C. truncatum*, *G. musarum*, and *P. calabae* MICs = 1.56–3.13 μg/mL	[99]
	116	Macrotermycin A	*Amycolatopsis* species M39	*C. albicans* ATCC 10231 MIC = 10 μg/mL *S. cerevisiae* ATCC9763 MIC = 5 μg/mL	[100]
	117	Macrotermycin C	*Amycolatopsis* species M39	*C. albicans* ATCC 10231 MIC = 25 μg/mL *S. cerevisiae* ATCC9763 MIC = 20 μg/mL	[100]
	118	F2928-1	fungus *Hypomyces pseudocorticiicola* FKA-73	*A. fumigatus* IFM 61493 MIC = 32 μg/mL *A. fumigatus* IFM 62104 MIC = 32 μg/mL *A. udagawae* IFM 62100 MIC = 32 μg/mL *A. felis* IFM 62093 MIC = 64 μg/mL *A. lentulus* IFM 62073 MIC = 64 μg/mL *C. auris* IFM 64524 MIC = 2 μg/mL *C. auris* IFM 65059 MIC = 2 μg/mL *C. auris* IFM 65061 MIC = 2 μg/mL	[101]
	119	Hakuhybotrol	fungus *Hypomyces pseudocorticiicola* FKA-73	*A. fumigatus* IFM 61493 MIC > 128 μg/mL *A. fumigatus* IFM 62104 MIC > 128 μg/mL *A. udagawae* IFM 62100 MIC > 128 μg/mL *A. felis* IFM 62093 MIC > 128 μg/mL *A. lentulus* IFM 62073 MIC > 128 μg/mL *C. auris* IFM 64524 MIC > 128 μg/mL *C. auris* IFM 65059 MIC > 128 μg/mL *C. auris* IFM 65061 MIC > 128 μg/mL	[101]
	120	Cladobotric acid A	fungus *Hypomyces pseudocorticiicola* FKA-73	*A. fumigatus* IFM 61493 MIC = 32 μg/mL *A. fumigatus* IFM 62104 MIC = 32 μg/mL *A. udagawae* IFM 62100 MIC = 32 μg/mL *A. felis* IFM 62093 MIC = 128 μg/mL *A. lentulus* IFM 62073 MIC = 64 μg/mL *C. auris* IFM 64524 MIC = 4 μg/mL *C. auris* IFM 65059 MIC = 8 μg/mL *C. auris* IFM 65061 MIC = 8 μg/mL	[101]
	121	Cladobotric acid F	fungus *Hypomyces pseudocorticiicola* FKA-73	*A. fumigatus* IFM 61493 MIC > 128 μg/mL *A. fumigatus* IFM 62104 MIC > 128 μg/mL *A. udagawae* IFM 62100 MIC > 128 μg/mL *A. felis* IFM 62093 MIC > 128 μg/mL *A. lentulus* IFM 62073 MIC > 128 μg/mL *C. auris* IFM 64524 MIC > 128 μg/mL *C. auris* IFM 65059 MIC > 128 μg/mL *C. auris* IFM 65061 MIC > 128 μg/mL	[101]
	122	Cladobotric acid E	fungus *Hypomyces pseudocorticiicola* FKA-73	*A. fumigatus* IFM 61493 MIC = 8 μg/mL *A. fumigatus* IFM 62104 MIC = 16 μg/mL *A. udagawae* IFM 62100 MIC = 32 μg/mL *A. felis* IFM 62093 MIC = 64 μg/mL *A. lentulus* IFM 62073 MIC = 32 μg/mL *C. auris* IFM 64524 MIC = 4 μg/mL *C. auris* IFM 65059 MIC = 2 μg/mL *C. auris* IFM 65061 MIC = 2 μg/mL	[101]
	123	Cladobotric acid H	fungus *Hypomyces pseudocorticiicola* FKA-73	*A. fumigatus* IFM 61493 MIC = 32 μg/mL *A. fumigatus* IFM 62104 MIC = 32 μg/mL *A. udagawae* IFM 62100 MIC = 64 μg/mL *A. felis* IFM 62093 MIC = 64 μg/mL *A. lentulus* IFM 62073 MIC = 64 μg/mL *C. auris* IFM 64524 MIC = 16 μg/mL *C. auris* IFM 65059 MIC = 16 μg/mL *C. auris* IFM 65061 MIC = 32 μg/mL	[101]
	124	Pyrenulic acid A	fungus *Hypomyces pseudocorticiicola* FKA-73	*A. fumigatus* IFM 61493 MIC = 32 μg/mL *A. fumigatus* IFM 62104 MIC = 32 μg/mL *A. udagawae* IFM 62100 MIC = 32 μg/mL *A. felis* IFM 62093 MIC > 128 μg/mL *A. lentulus* IFM 62073 MIC > 128 μg/mL *C. auris* IFM 64524 MIC = 16 μg/mL *C. auris* IFM 65059 MIC = 16 μg/mL *C. auris* IFM 65061 MIC = 16 μg/mL	[101]
	125	Campafungin A	*Plenodomus enteroleucus* Strain F-146,176	*C. neoformans* H77 MIC = 8 μg/mL	[102]
	126	Campafungin B	*Plenodomus enteroleucus* Strain F-146,176	*C. neoformans* H77 MIC = 4 μg/mL	[102]
	127	Campafungin C	*Plenodomus enteroleucus* Strain F-146,176	*C. neoformans* H77 MIC = 4 μg/mL	[102]
	128	Campafungin D	*Plenodomus enteroleucus* Strain F-146,176	*C. neoformans* H77 MIC = 8 μg/mL	[102]

## 9. Conclusions

In conclusion, this review provides a comprehensive overview of natural antifungal polyketides encompassing various subclasses such as polyethers, macrolides, xanthones, linear polyketides, anthraquinone, polyphenols, pyridine derivatives, furan derivatives, pyranan derivatives, monophenyl derivatives, macrolactam polyketides, hybrid polyketide non-ribosomal peptides, and other polyketides. Additionally, this review discusses the origin, in vitro and in vivo antifungal activities, structure–activity relationship (SAR), safety profile, mechanism of action, and the impact of structural modifications on the SAR of these polyketides. Previous studies on polyketides have demonstrated the substantial antifungal properties exhibited by certain natural polyketides, such as amphotericin B and caspofungin. This observation suggests that polyketide lead compounds hold considerable potential for the future treatment of fungal infections. Given the remarkable antifungal activities displayed by natural polyketides, this class of compounds has garnered significant interest as a potential therapeutic avenue for fungal infections in the future. Currently, the predominant research emphasis lies in synthesizing novel polyketides, while the clinical utilization of pre-existing polyketide compounds as antifungal medications remains limited. Consequently, further comprehensive and meticulous clinical investigations are imperative to substantiate their efficacy in the future. 

Moreover, the antifungal properties of unnatural polyketide compounds can potentially be harnessed through combinatorial biosynthesis. Numerous PKSs facilitate the production of primary polyketide compounds, which lack biological activity until they undergo modification by PKS post-modifying enzymes, thereby presenting a promising avenue for exploring new antifungal drugs. Polyketides’ chemical composition and fungicidal properties can be altered by utilizing various post-modification enzymes, including cyclase, aromatase, glycosylase, and halogenase. Unconventional antifungal polyketides have been synthesized by modifying the modules, domains, and subunits of PKSs and employing site-directed mutagenesis techniques. Furthermore, the enhancement of antifungal polyketide production or the acquisition of novel antifungal polyketides can be achieved by combining initiation substrates and elongation units from different hosts and implementing targeted modifications. Only a small number of antifungal natural polyketide compounds have been identified. Many habitats of microorganisms, plants, animals and marine organisms have not been explored, and many antifungal polyketide products urgently need to be discovered. Natural polyketides from microorganisms such as fungi are scarce and difficult to obtain. The solubility, safety, and in vivo bioavailability of natural polyketides should also be considered. Semisynthetic components of natural products will play an important role in developing antifungal candidates in the future.

## Figures and Tables

**Figure 1 biomolecules-13-01572-f001:**
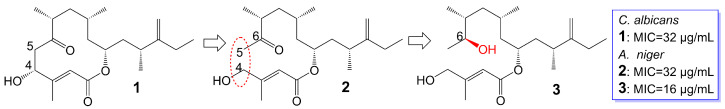
Chemical structures of Amphidinins Q (**1**), C (**2**), and E (**3**). The red dotted box marks the open-loop structure.

**Figure 2 biomolecules-13-01572-f002:**
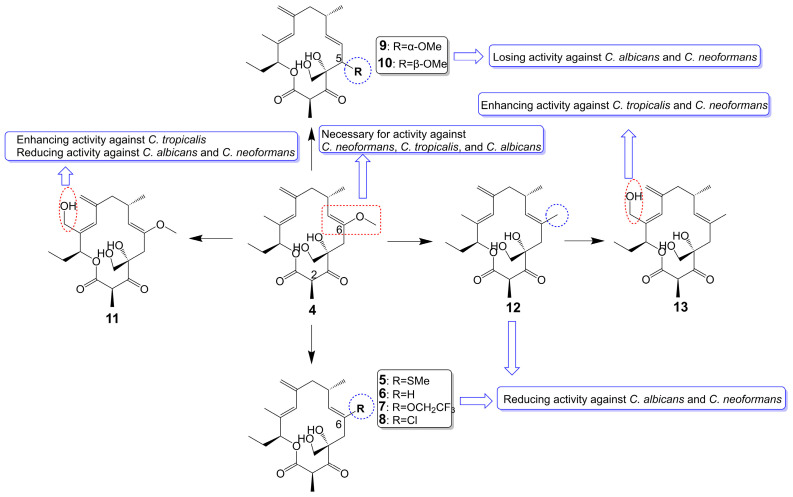
Chemical structures of rustmicin (**4**) and its analogues.

**Figure 3 biomolecules-13-01572-f003:**
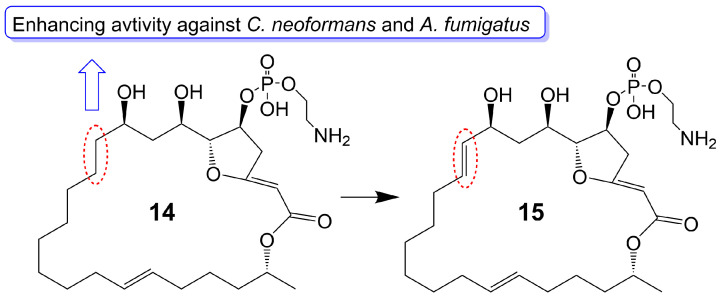
Chemical structures of Preussolides A (**14**) and B (**15**). The red dotted boxes mark the difference between Preussolides A (**14**) and B (**15**).

**Figure 4 biomolecules-13-01572-f004:**
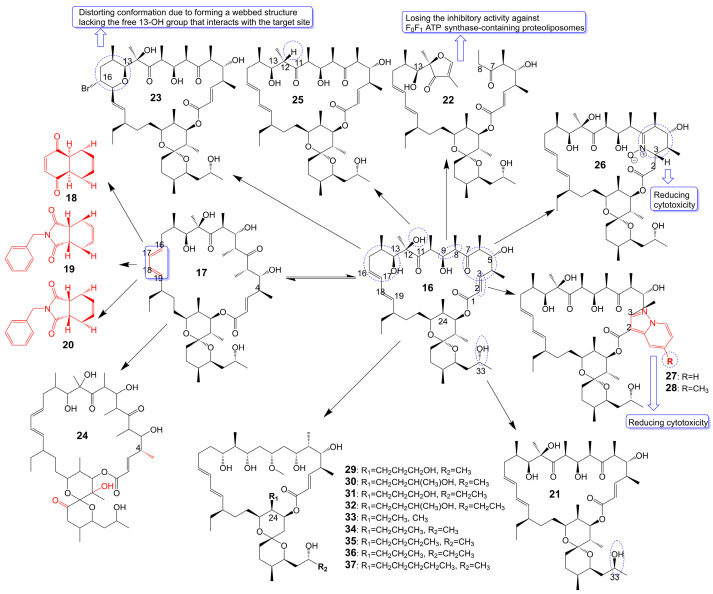
Chemical structures of analogues of Oligomycin A.

**Figure 5 biomolecules-13-01572-f005:**
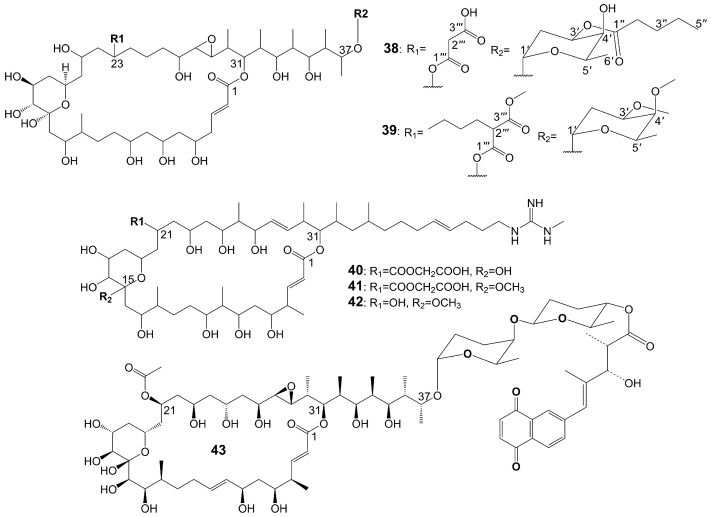
Chemical structures of Brasilinolides A (**38**) and B (**39**) and their analogues.

**Figure 6 biomolecules-13-01572-f006:**
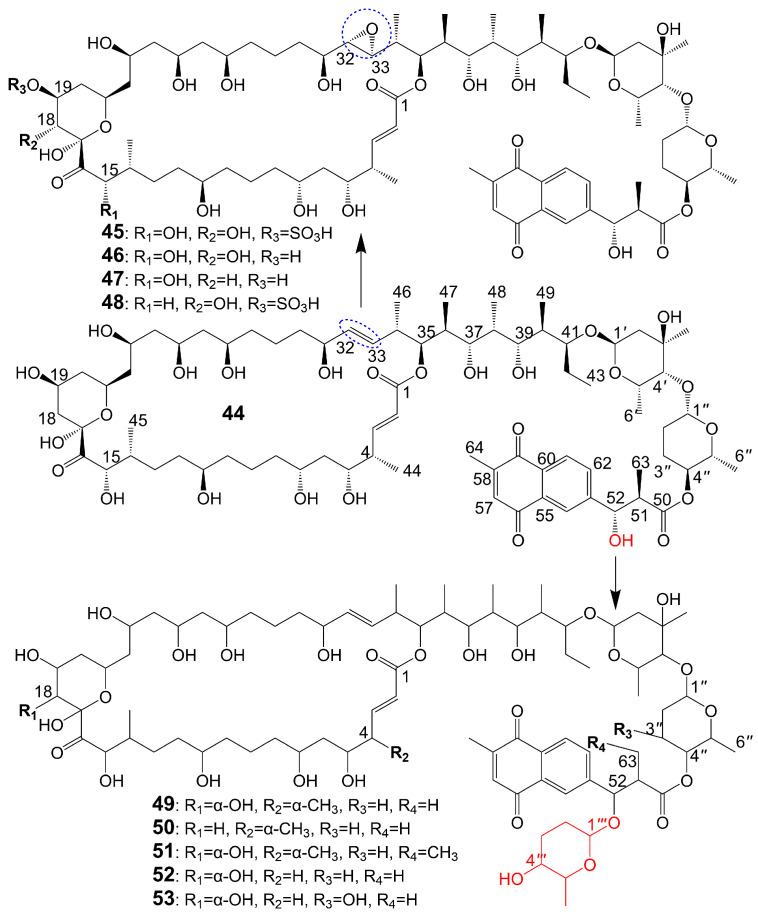
Chemical structures of cyphomycin (**44**) and its analogues. The blue dotted boxs mark the difference between cyphomycin (**44**) and caniferolide C (**47**).

**Figure 7 biomolecules-13-01572-f007:**
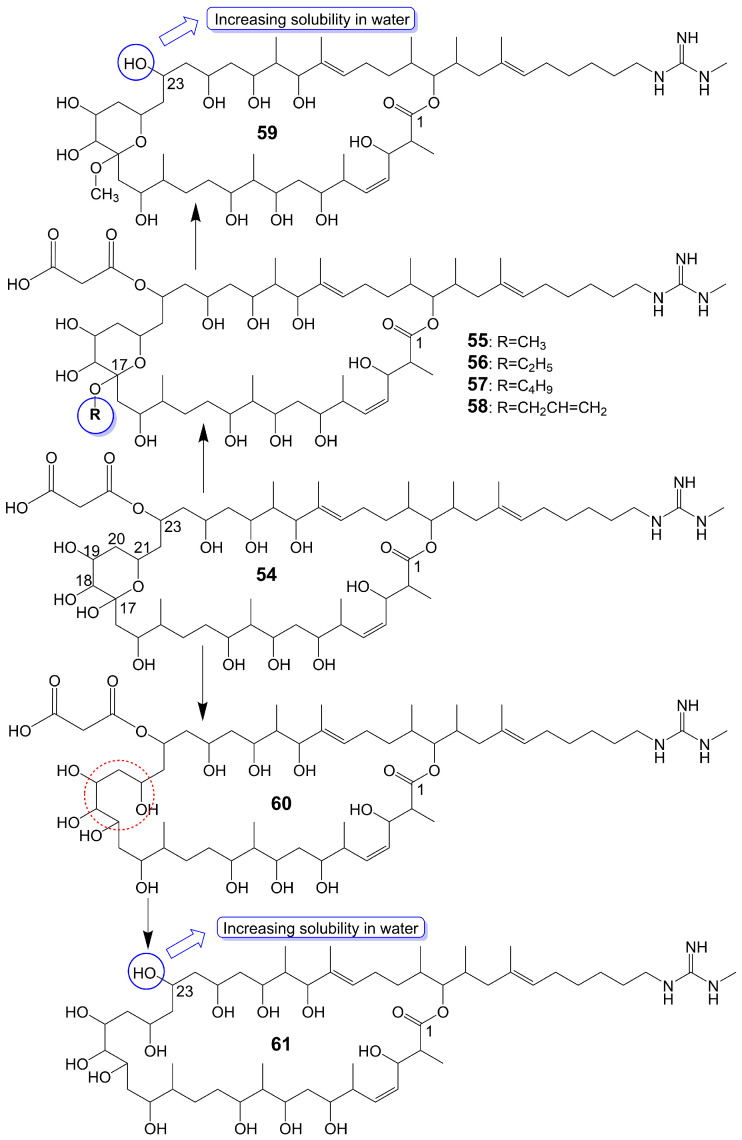
Chemical structures of Guanidylfungin A (**54**) and its analogues. The red dotted box marks the ring-opening structure of compound **60**, which differs from the tetrahydropyran ring guanidylfungin A (**54**).

**Figure 8 biomolecules-13-01572-f008:**
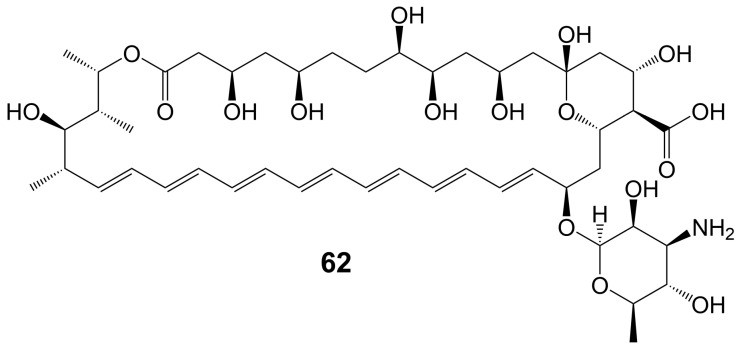
Chemical structures of amphotericin B (**62**).

**Figure 9 biomolecules-13-01572-f009:**
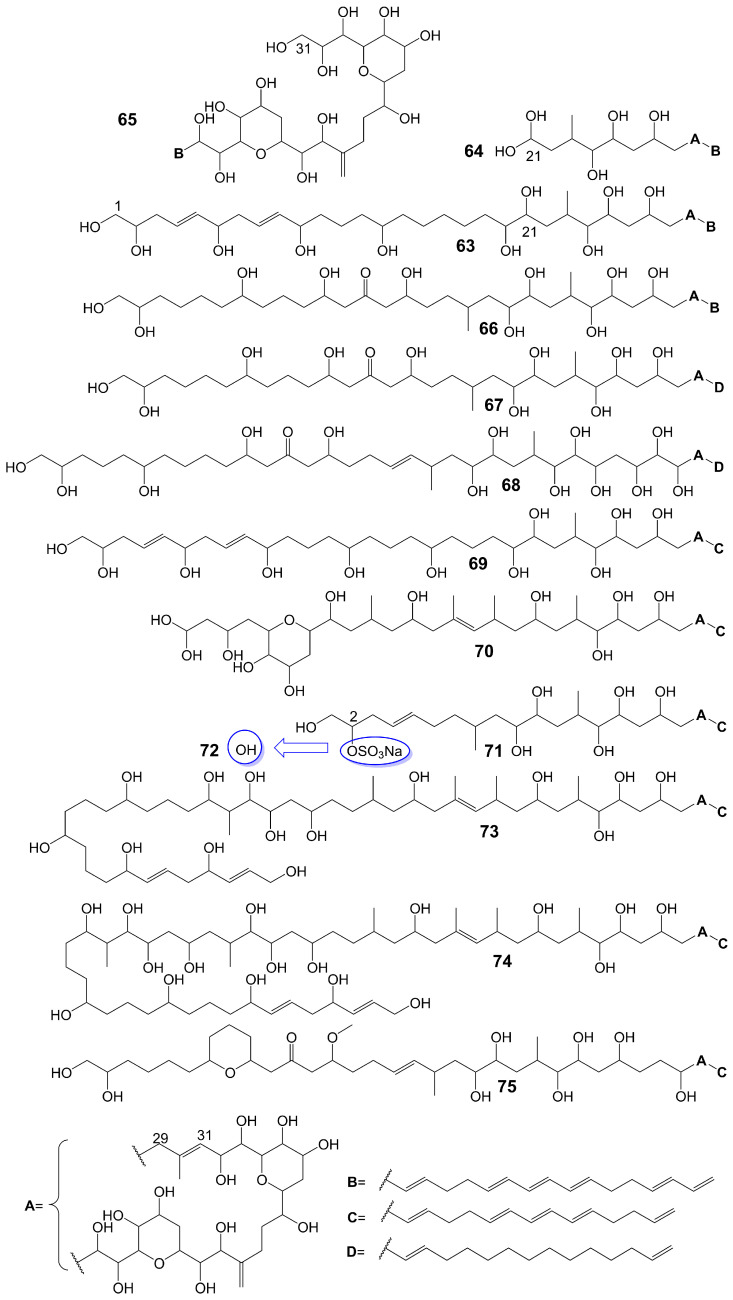
Chemical structures of amphidinols.

**Figure 10 biomolecules-13-01572-f010:**
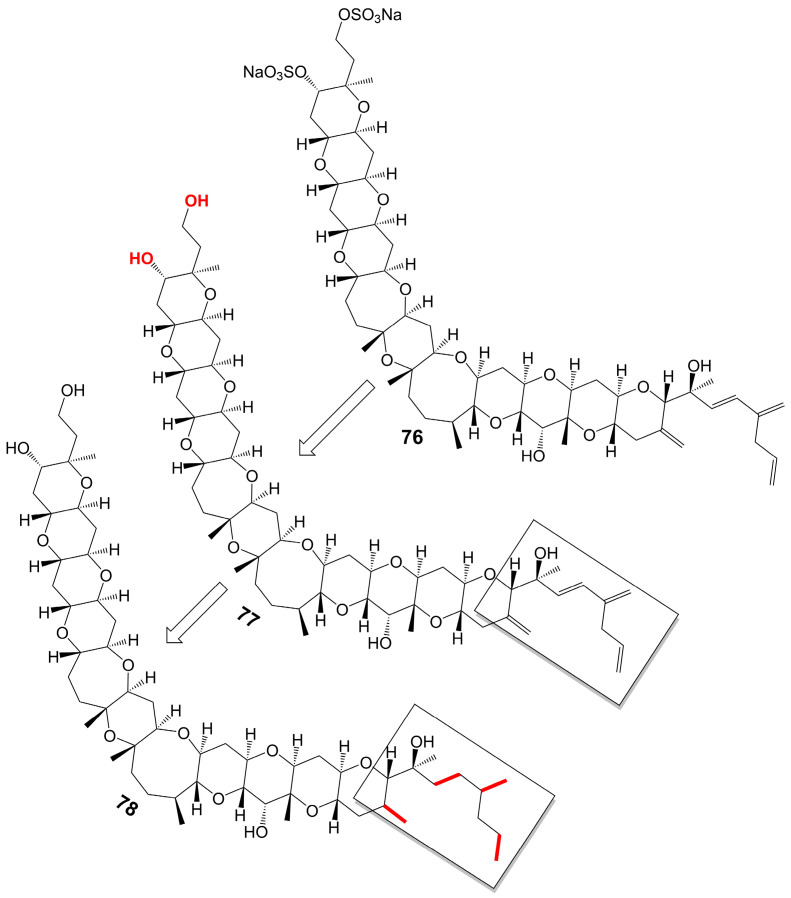
Chemical structures of yessotoxin (**76**), desulfated yessotoxin (**77**), and hydrogen-desulfated yessotoxin (**78**).

**Figure 11 biomolecules-13-01572-f011:**
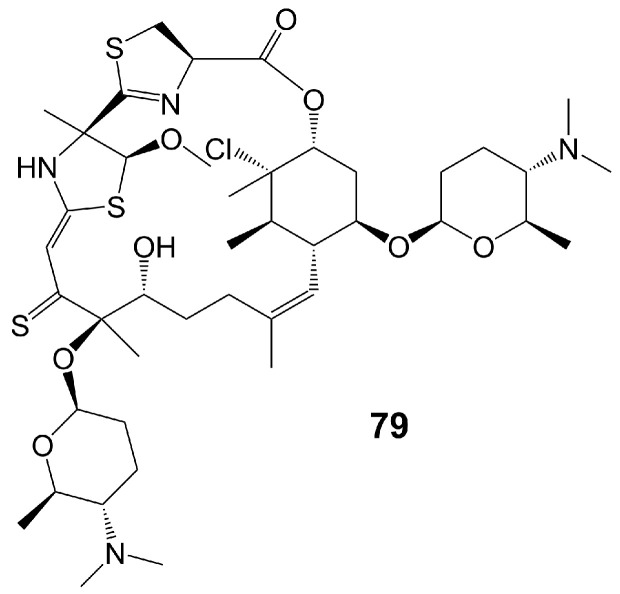
Chemical structures of forazoline A (**79**).

**Figure 12 biomolecules-13-01572-f012:**
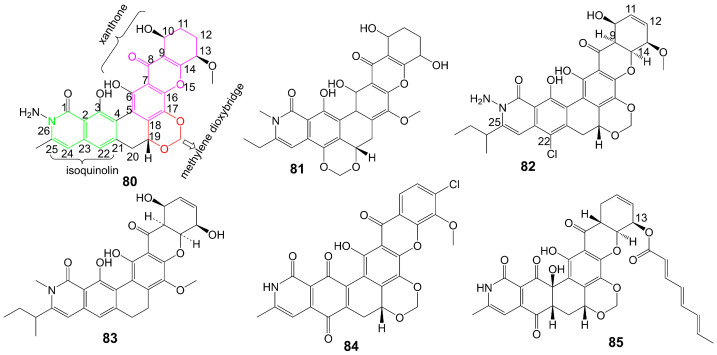
Chemical structures of albofungin (**80**) and its analogues.

**Figure 13 biomolecules-13-01572-f013:**
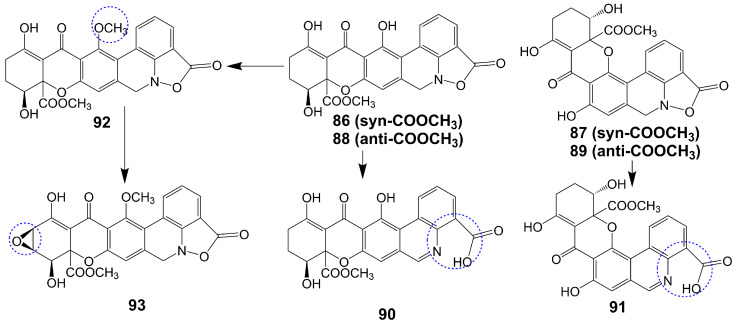
Chemical structures of xanthone polyketides parnafungins A1 (**86**) and B1 (**87**) and their analogues.

**Figure 14 biomolecules-13-01572-f014:**
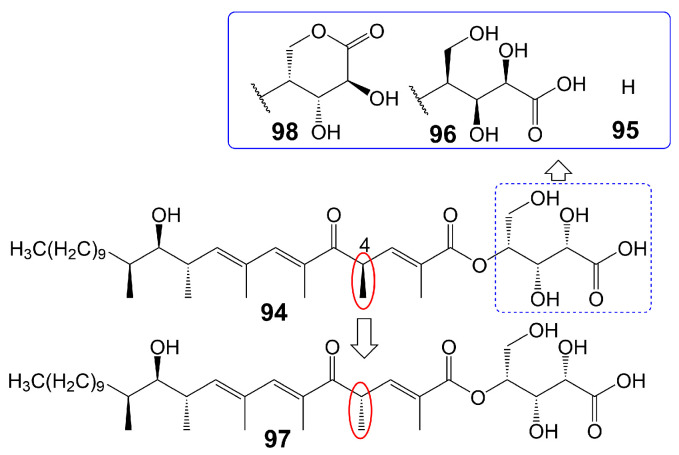
Chemical structures of khafrefungin (**94**) and its analogues. The red boxes mark the difference between khafrefungin (**94**) and **97** of the 4-methyl group.

**Figure 15 biomolecules-13-01572-f015:**
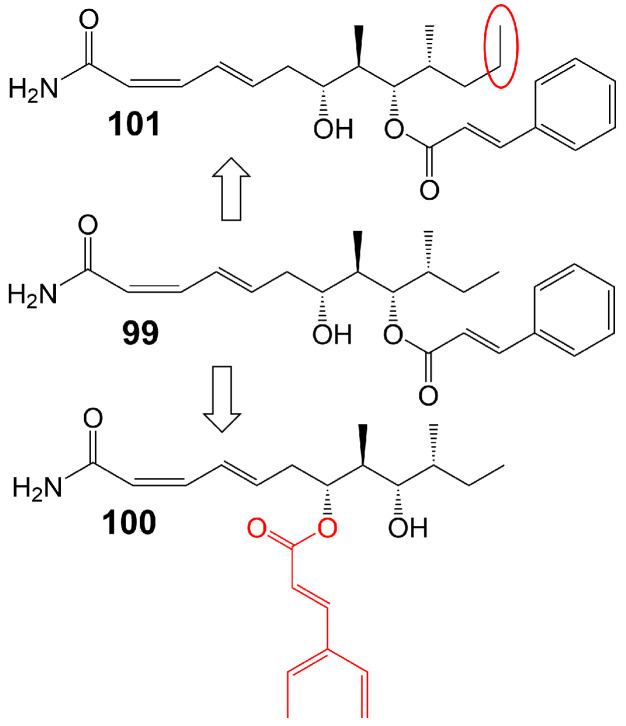
Chemical structures of basiliskamide A (**99**) and its analogues. The red box marks an additional methylene group of YM-45722 (**101**), which differs from basiliskamide A (**99**).

**Figure 16 biomolecules-13-01572-f016:**
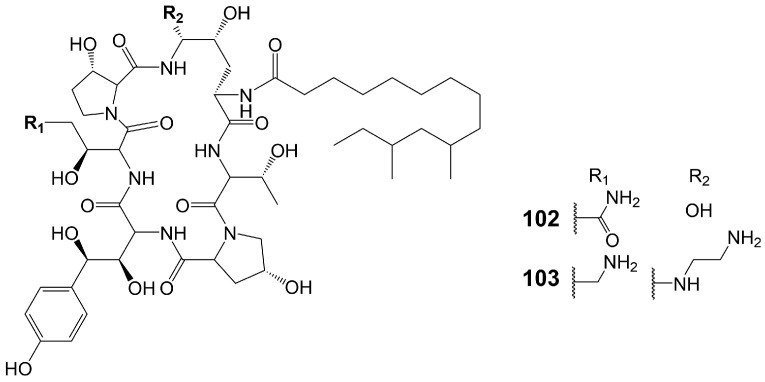
Chemical structures of hybrid polyketide nonribosomal peptides.

**Figure 17 biomolecules-13-01572-f017:**
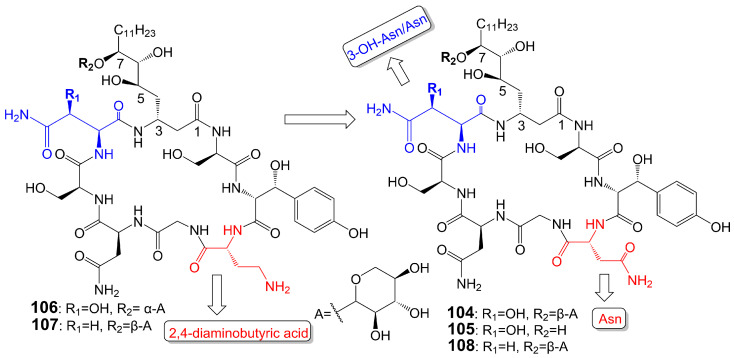
Chemical structures of burkholdines 1229 (**104**) and its analogues.

**Figure 18 biomolecules-13-01572-f018:**
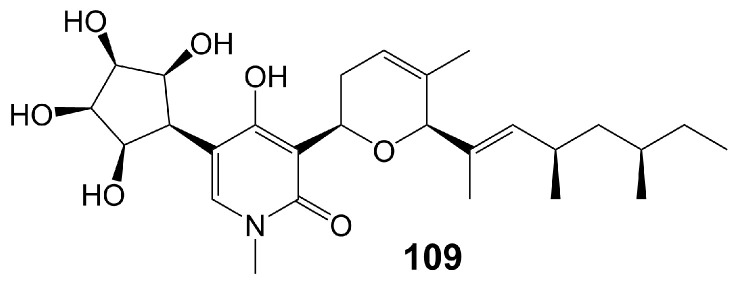
Chemical structures of funiculosin (**109**).

**Figure 19 biomolecules-13-01572-f019:**
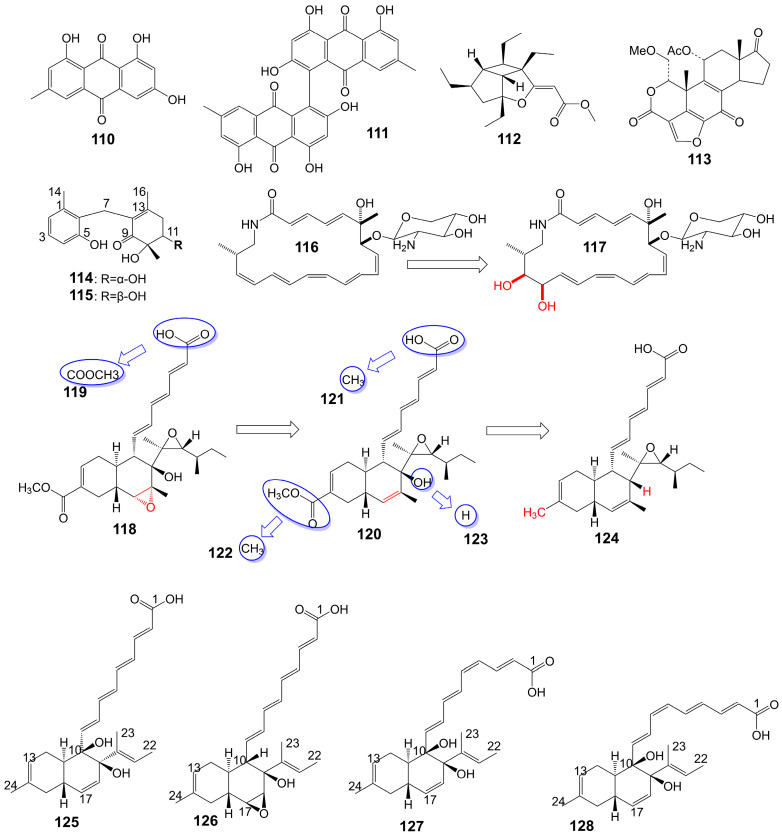
Chemical structures of other polyketides.

## Data Availability

The data presented in this study are available in this article.

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
