# Peer review of "Natural Polyketides Act as Promising Antifungal Agents"

_biomolecules, 2023, doi:10.3390/biom13111572_

Round 1

Reviewer 1 Report

The manuscript by Li Wang et al. reviews natural polyketides with an assessment of their mechanism of action and minimum doses. Overall, the authors presented an interesting review that is consistent with the chosen topic of the Biomolecules journal.

However, the manuscript contains a number of typographical errors that need to be corrected. For example, the Latin names of bacteria should be highlighted throughout the text.

Is line 118 "albonolid B" a new substance that the authors are describing, or is this a typo and "galbonolid B" is the correct name?

Line 144: "niger" must start with a lowercase letter

Lines 136, 151 and 156 should be given the same designation s-trans and s-cis.

Line 637. The “Discussion and Conclusion” section should be renamed “Conclusion”, since in this section the authors summarize the presented review. In fact, there is no discussion here.

One final note. I, of course, am not a native English speaker and it is difficult for me to judge the legitimacy of the grammatical structures used by the authors, but it seems to me that English should be checked by a native speaker. In my opinion, the Authors in a number of cases use the active form of the predicate where there should be a passive.

For example, lines 217-218. It is written that "Caniferolide D, in contrast to caniferolide A, removes the hydroxyl group of C-15 position." Most likely the correct sentense is “Caniferolide D, unlike kaniferolide A, has no hydroxyl group at position C-15.” Is it so?

Lines 225-226. . It is written that “Iseolide C, which attaches a methyl group at position C-63 of the glycosylated portion of iseolide A, has greatly reduced antifungal activity….” I would say that that Iseolide C itself does not attach to anything, but contains this methyl group.

Similar remarks to the grammar of the sentences on lines 229-230, 282-283 (“undergoes” or “is undergone”?), lines 394-395.

Line 42. “There compounds” - what compounds are we talking about? Polyketides or “acetyl-CoA, propio-41 nyl-CoA, malonyl-CoA, and methylmalonyl-CoA”, which are the last ones mentioned?

Overall, I found this review very interesting and can be recommended for publication.

Reviewer 2 Report

The review is carefully prepared and contains a number of supporting references. Information is basically provided in the format: Name, Structure, Source, and Activity. However, polyketides are not  new class of compounds, which would be "promising", but number of them are already used, there are in some stage of clinical trials or approval process. With this respect, the MIC value only does not enable to distinguish, what is "promising" and what is already approved drug. The problem is that there are already published reviews, which provide such comparison in tables, or are more deeply dedicated like "Spirotetronate polyketides as leads in drug discovery.

Reviewer 3 Report

RAS

Round 2

Reviewer 2 Report

Manuscript is now well organised and Tables are veryhelpful.

Reviewer 3 Report

The authors have adress the majority of my comments. 

The quality of the paper has greatly increased. It could be publish now. 

No particular comment on English language.